# Mechanisms and functional roles of glutamatergic synapse diversity in a cerebellar circuit

Valeria Zampini[1,2,3†], Jian K Liu[4,5,6†], Marco A Diana[1,2,3‡],
Paloma P Maldonado[1,2,3§], Nicolas Brunel[4,7*], Stéphane Dieudonné[1,2,3*]

[1]Institut de Biologie de l'ENS, Ecole Normale Supérieure, Paris, France; [2]Inserm, U1024, Paris, France; [3]CNRS, UMR 8197, Paris, France; [4]Neurosciences Federation, Université Paris Descartes, Paris, France; [5]Department of Ophthalmology, University Medical Center Goettingen, Goettingen, Germany; [6]Bernstein Center for Computational Neuroscience, Göttingen, Germany; [7]Department of Statistics and Neurobiology, University of Chicago, Chicago, United States

*For correspondence: nbrunel@ uchicago.edu (NB); dieudon@ biologie.ens.fr (SD)

[†]These authors contributed equally to this work

Present address: [‡]Sorbonne Universités, UPMC Univ Paris 06, INSERM, CNRS, Neurosciences Paris Seine - Institut de Biologie Paris Seine (NPS - IBPS), Paris, France; [§]Netherlands Institute for Neuroscience, the Royal Academy of Arts and Sciences, Amsterdam, the Netherlands

Competing interests: The authors declare that no competing interests exist.

**Abstract** Synaptic currents display a large degree of heterogeneity of their temporal characteristics, but the functional role of such heterogeneities remains unknown. We investigated in rat cerebellar slices synaptic currents in Unipolar Brush Cells (UBCs), which generate intrinsic mossy fibers relaying vestibular inputs to the cerebellar cortex. We show that UBCs respond to sinusoidal modulations of their sensory input with heterogeneous amplitudes and phase shifts. Experiments and modeling indicate that this variability results both from the kinetics of synaptic glutamate transients and from the diversity of postsynaptic receptors. While phase inversion is produced by an mGluR2-activated outward conductance in OFF-UBCs, the phase delay of ON UBCs is caused by a late rebound current resulting from AMPAR recovery from desensitization. Granular layer network modeling indicates that phase dispersion of UBC responses generates diverse phase coding in the granule cell population, allowing climbing-fiber-driven Purkinje cell learning at arbitrary phases of the vestibular input.

## Introduction

Sensory stimuli are encoded by populations of neurons with a diversity of spatio-temporal selectivity properties, which allow sensory systems to obtain accurate representations of such stimuli over the whole range of ecological spatial and temporal time scales. In the vestibular system, primary vestibular neurons primarily encode the rotational velocity (semicircular canals) and linear acceleration (gravitational or inertial, in the otholitic organs) of the head (*Arenz et al., 2008*; *Fernandez and Goldberg, 1971*; *Goldberg, 2000*; *Goldberg and Fernandez, 1971a*, *1971b*; *Green and Angelaki, 2010*; *Sadeghi et al., 2007*). These vestibular inputs are in turn used to adapt body posture, generate compensatory and orienting eye movements, and modify vasomotor autonomic functions.

The vestibulo-cerebellum, which is innervated by primary vestibular afferents in the vermal part (*Barmack et al., 1993*; *Chabrol et al., 2015*; *Gerrits et al., 1989*; *Korte and Mugnaini, 1979*) and by secondary vestibular afferents in all its subsections (*Barmack et al., 1992*; *Magras and Voogd, 1985*; *Thunnissen et al., 1989*), is well known to perform the sensory-motor integration required to orient the body in space and to generate accurate eye movements. Computational models have suggested that, in order to perform such functions, vestibular inputs should be pre-processed by filters with a wide diversity of temporal scales so that learning can take place at synapses onto Purkinje cells (*Dean et al., 2010*; *Fujita, 1982*). A similar idea was proposed for suppression of the self-

**eLife digest** Whether walking, riding a bicycle or simply standing still, we continually adjust our posture in small ways to prevent ourselves from falling. Our sense of balance depends on a set of structures inside the inner ear called the vestibular system. These structures detect movements of the head and relay this information to the brain in the form of electrical signals. A brain area called the vestibulo-cerebellum then combines these signals with sensory input from the eyes and muscles, before sending out further signals to trigger any adjustments necessary for balance.

One of the main cell types within the vestibulo-cerebellum is the unipolar brush cell (or UBC for short). UBCs pass on signals to another type of neuron called Purkinje cells, which support the learning of motor skills such as adjusting posture. Zampini, Liu et al. set out to test the idea that UBCs transform inputs from the vestibular system into a format that makes it easier for cerebellar Purkinje cells to drive this kind of learning.

First, recordings from slices of rodent brain revealed that UBCs respond in highly variable ways to vestibular input, with both the size and timing of responses varying between cells. This is because vestibular signals trigger the release of a chemical messenger called glutamate onto UBCs, but UBCs possess a variety of different types of glutamate receptors. Vestibular input therefore activates distinct signaling cascades from one UBC to the next. According to a computer model, this variability in UBC responses ensures that a subset of UBCs will always be active at any point during vestibular input. This in turn means that Purkinje cells can fire at any stage of a movement, which boosts the learning of motor skills.

The next steps will be to test this hypothesis using mutant mice that lack specific receptor subtypes in UBCs or UBCs completely. A further challenge for the future will be to build a computer model of the vestibulo-cerebellar system that includes all of its component cell types.

generated sensory signal in the electric fish (*Kennedy et al., 2014*; *Roberts and Bell, 2000*). Despite the central importance of this filtering process for vestibulo-cerebellar function, it remains unclear at which stage and through which underlying mechanisms this pre-processing takes place. In vivo electrophysiological recordings during passive head rotation indicate that primary vestibular mossy fibers activity displays a high level of stereotypy with mostly in-phase firing during sinusoidal velocity modulations (*Arenz et al., 2008*). Filtering may nevertheless occur upstream of granule cells, as the available in vivo recordings of granule cells show a large diversity of phase shifts in response to head rotations (*Barmack and Yakhnitsa, 2008*). This suggests that pre-processing occurs in the granular layer, as recently suggested in the electric fish electrosensory lobe (*Kennedy et al., 2014*).

Extrinsic vestibular extrinsic mossy fibers (eMFs) project to granule cells but also to another cell type, the unipolar brush cells (UBCs). Each UBC receives a single mossy fiber input, and projects onto granule cells by forming several intrinsic mossy fiber terminals (iMFs). The single mossy fiber contacting the dendritic brush of UBCs forms a giant synapse of unique morphology (*Mugnaini et al., 1994*) at which various types of glutamate receptors are expressed (*Jaarsma et al., 1995*). Entrapment of glutamate in the synaptic cleft can lead to responses on slow time scales (*Kinney et al., 1997*; *van Dorp and De Zeeuw, 2014*). UBCs have been divided into different sub-types by a number of studies (*Borges-Merjane and Trussell, 2015*; *Dino et al., 1999*; *Mugnaini et al., 1997*; *Nunzi et al., 2002*). In particular, *Borges-Merjane and Trussell (2015)* showed recently that dorsal cochlear nucleus UBCs can be divided into ON UBCs, which have an excitatory response to their mossy fiber inputs mediated by both AMPA and mGluR1 receptors, and into OFF UBCs, the response of which is inhibitory and is produced by mGluR2 receptor activation.

These studies still leave unanswered the question whether and how UBC pre-processing generates the diversity of tuning properties of granule cells. To answer this question, we first set out to characterize the input/output relationship of UBCs in the presence of sinusoidal MF stimulation. Our experiments show that UBCs have a wide diversity of tuning properties, due to the diversity of their synaptic responses. We then investigated the functional consequences of this diversity on the computational properties of the vestibulocerebellar circuit, using a network model. We demonstrate

that UBCs, through the generation of diverse response profiles, greatly enhance the ability of Purkinje cells to learn arbitrary input/output mappings.

## Results

### Responses to steady MF stimulations defines OFF and ON UBCs

In vivo, afferent vestibular inputs fire steadily at mean rates of 10–50 Hz at rest (*Arenz et al., 2008*; *Barmack and Yakhnitsa, 2008*; *Dickman et al., 1991*; *Fernandez and Goldberg, 1971*; *Goldberg, 2000*; *Goldberg and Fernandez, 1971a*, *1971b*; *Green and Angelaki, 2010*; *Loe et al., 1973*; *Sadeghi et al., 2007*; *Tomko et al., 1981*). We first examined the impact of a continuous MF input on the discharge rate of UBCs in acute brain slices using a typical firing rate of 26 Hz. GABAergic and glycinergic components were blocked during the experiments by perfusion of SR95531 (2 µM), CGP 55,845 (0.5 mM), and strychnine (1 µM). This simple protocol evoked three types of behavior. In a first group of cells (OFF-UBCs, n= 25, *Figure 1A*), the glutamatergic MF input evoked a hyperpolarization ($-9.5 \pm 4.2$ mV at steady-state). In 5 cells the hyperpolarization was preceded by a transient period of action potential firing ($4 \pm 2$ spikes at $26.3 \pm 15.2$ Hz, duration $0.12 \pm 0.06$ s). Out of 25 OFF-UBCs, 6 fired spontaneously ($12.8 \pm 8.1$ Hz) at rest. MF stimulation silenced 5 of these UBCs and decreased firing rate by 28% in the last one.

In a second group of cells (ON-UBCs, n= 22, *Figure 1F*) MF synaptic inputs depolarized the neuron, as expected from a glutamatergic fiber. At steady-state ON-UBCs were depolarized by $10.4 \pm 5.9$ mV from a resting membrane potential of $-60.7 \pm 3.7$ mV and fired at $11.7 \pm 9.2$ Hz with an interspike interval $CV_2$ of $0.435 \pm 0.239$ (see Experimental Procedures). Overall, ON-UBCs covered a range of frequencies ($1 - 29.9$ Hz) similar to that found in vivo under anesthesia (*Barmack and Yakhnitsa, 2008*; *Ruigrok et al., 2011*; *Simpson et al., 2005*). Only 1 of 22 ON UBCs fired at rest (1.7 Hz) in the cell-attached configuration.

Finally, a group of complex UBCs (n = 10) displayed delayed or gradual depolarization which did not reach steady-state at the end of the MF stimulation protocol (*Figure 1—source data 1A,E*). Similarly to OFF-UBCs, 8 of these 10 UBCs first hyperpolarized and then resumed firing at $10.9 \pm 10.5$ Hz with a delay from stimulation onset of $1.2 \pm 1.3$ s (from 0.44 to 4.16 s). The slow inward current underlying the depolarization (*Figure 1—source data 1B*) was blocked by the group I mGluR antagonist JNJ16259685 (*Figure 1—source data 1C*). In some cells the slow depolarization persisted after MF stimulation was stopped, leading to several seconds (min 2 s, max >10 s) of firing ($5.6 \pm 3.7$ Hz, min 3.3; max 10.9). These complex UBCs behave as the very low frequency integrators predicted by vestibular psychophysics (*Green and Angelaki, 2010*). They could nonetheless not be studied here within the timescale of our stimulation protocols and were discarded from further experiments.

### OFF-UBCs and ON-UBCs respond to sinusoidal modulation with different phase shifts and bursting profiles

OFF-UBCs and ON-UBCs were further exposed to 1 Hz modulations of their MF input rate (*Figures 1A and F*) (see Experimental Procedures), as seen for vestibular MFs during sinusoidal head rotations in vivo (*Arenz et al., 2008*; *Fernandez and Goldberg, 1971*). OFF- and ON-UBCs responded to MF rate modulation with phase-locked firing (*Figures 1B and G*), as assessed by fitting the phase distribution of firing rates with a circular normal function,

$$f(\theta) = r_{min} + (r_{max} - r_{min})\left(e^{k^2cos(\theta-\phi)} - e^{-k^2}\right)/\left(e^{k^2} - e^{-k^2}\right),$$

(see Experimental Procedures), where $r_{max}$ ($r_{min}$) is the maximal (minimal) firing rate, $\phi$ is the preferred phase shift and $k$ is a measure of concentration of the spike phases (n = 47) (*Figures 1C and H*). A striking feature of the UBC responses was the broad diversity of their phase shift (*Figures 1D and I*) relative to the peak of the MF stimulation (*Figures 1B and G*, blue arrow). The phase shift of ON-UBCs was broadly distributed (SD of 105°; *Figure 1I*) and covered essentially all possible phase shifts, with a sum of phase preferences weighted by the firing modulation strength of 140°. OFF-UBCs fired within a narrower range of phases ($274 \pm 28°$; *Figure 1D*) and had a significantly higher phase lag than ON-UBCs (p<0.001; *Figure 1J*). During 1 Hz modulation 4 OFF-UBCs were silenced at all phases.

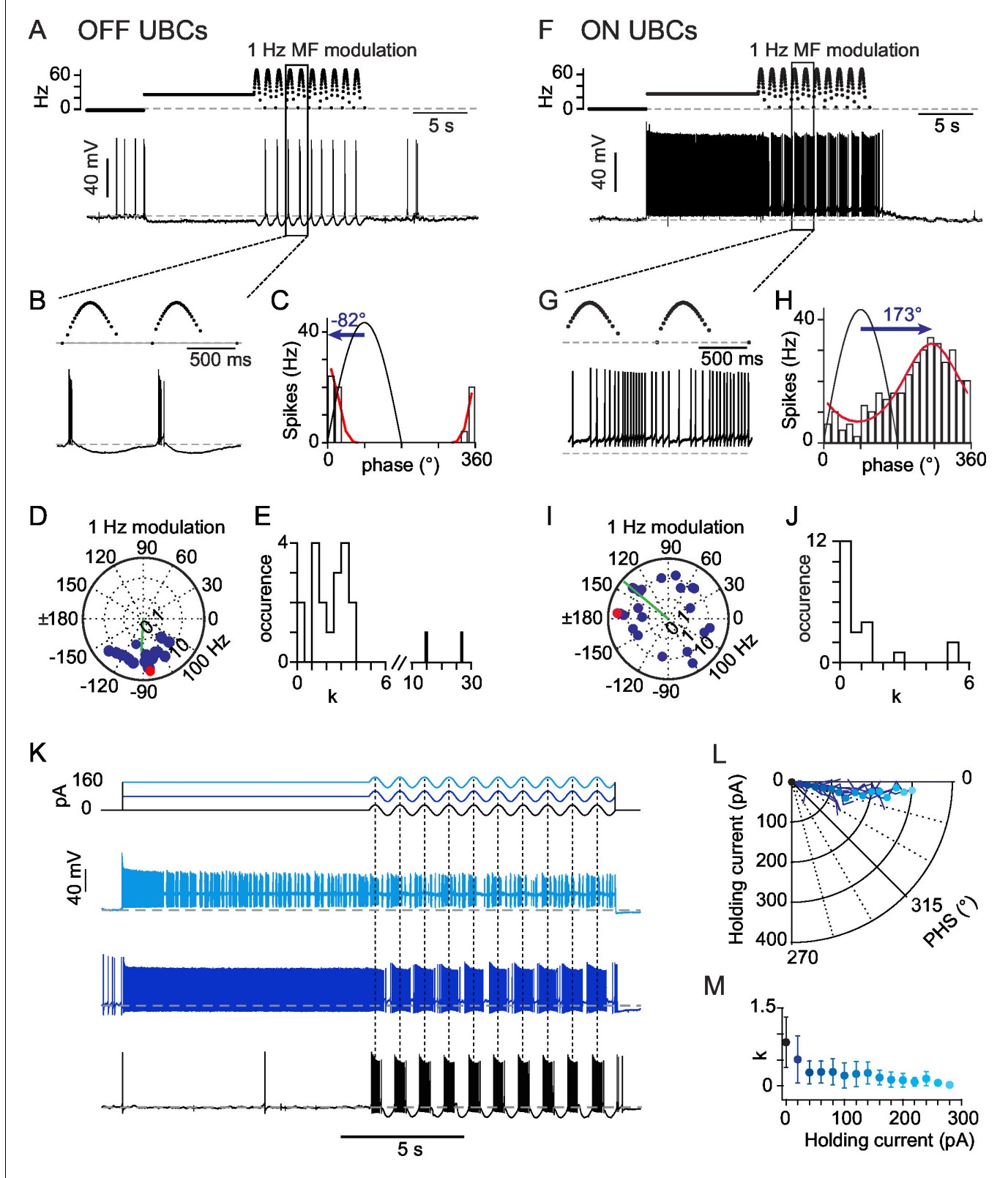

**Figure 1.** OFF- and ON- responses to MF stimulations in two classes of UBCs. (**A**) Current-clamp response of a *OFF-UBC* to a prolonged stimulation of its afferent MF at 26 Hz showing the OFF hyperpolarizing behavior. Steady stimulation is followed by a 1 Hz sinusoidal modulation of the MF stimulation rate. (**B**) Enlargement of the selected region in (**A**) showing the phase of UBC spiking relative to the MF stimulation modulation. (**C**) Fit of the *OFF-UBC* instantaneous firing rate vs phase relationship, obtained with the ten stimulation cycles shown in (**A**), using an exponentiated cosine

*Figure 1 continued on next page*

*Figure 1 continued*

function (see Materials and methods) Here and in the next figures the MF stimulation frequency is depicted in black and peaks at 90°. The blue arrow represents the UBC phase shift relative to the MF stimulation. (D) Polar plot of the phase shift and firing frequency modulation of *OFF-UBCs* (n = 20) obtained with 1 Hz sinusoidal modulations of the MF stimulation rate. The cell in (A) is plotted in red. (E) Distribution of the concentration factor $k^{1/2}$ yielded by the exponentiated cosine fit of *OFF-UBC* responses in (C) (n = 20). (F–J) Same as (A–E) for *ON-UBCs*. (K) Typical UBC current-clamp responses to the current injections displayed on top traces, where a sinusoidal current modulation of constant amplitude is superimposed on a variable holding current. (L) Phase shift of the UBC responses to sinusoidal current injections shown in (K) as a function of the injected holding current. (M) $k^{1/2}$ values of the UBC responses to sinusoidal current injections shown in (K) as a function of the injected holding current.

The following source data is available for figure 1:

**Source data 1.** Numerical data corresponding to panels D, E, I, J, L, M of *Figure 1*.

The firing modulation amplitude of OFF and ON cells was similar (12 ± 8 and 12 ± 23 Hz; respectively p=0.353) but the dispersion of action potentials along the modulation cycle differed between these two populations. We obtained the spike phase concentration factor $k$ from the fit to the phase histogram. OFF-UBCs fired in short bouts (*Figure 1E*; $k$ = 2.2 ± 1.2; 16 out of 18 cells with $k$>1). In contrast most ON-UBCs displayed a near-sinusoidal behavior (*Figure 1J*; $k$ = 0.9 ± 1.5; 15 out of 22 cells with $k$<1; p=0.0006), with only 7 cells displaying somewhat clustered responses ($\sqrt{k}$>1; mean 2.2). Thus, OFF and ON UBCs respond to time-modulated MF inputs with different phase-modulated firing behaviors and non-linearity.

UBCs are endowed with a complex set of active membrane conductances which may influence their phase response (*Afshari et al., 2004*; *Diana et al., 2007*; *Locatelli et al., 2013*; *Russo et al., 2007*). To quantify the effects of these intrinsic conductances we recorded the response of UBCs to sinusoidal current injections at 1 Hz (± 30 pA) (*Figure 1K*), while a holding current bias was varied to mimic the tonic drive occurring in ON and OFF UBCs. We found that all UBCs (n = 21) responded with a modest phase advance of 9.9 ± 4.1° (*Figure 1L*), independently of the holding current bias, up to the sodium spike inactivation threshold (200 ± 85 pA). The sharpness of the UBC response was strongly regulated by the holding current, with hyperpolarized cells firing spikes concentrated at the peak of the sinusoidal current injection (*Figure 1M*). This effect is likely to account for the difference of clustering factor between ON and OFF UBCs (*Figures 1E and J*). Hence, the strong heterogeneity in UBC phase shifts in response to MF stimulation is not due to intrinsic conductances. We therefore turned our attention to synaptic conductances to explain this heterogeneity.

## mGluR2 mediated hyperpolarization of OFF-UBCs implements phase reversion

A subpopulation of UBCs is known to express group II mGluRs (mGluR2) (*Borges-Merjane and Trussell, 2015*; *Jaarsma et al., 1998*; *Rousseau et al., 2012*) which are positively coupled to a GIRK2 containing inward rectifying potassium channel (*Knoflach and Kemp, 1998*; *Rousseau et al., 2012*; *Russo et al., 2007*). Synaptic currents evoked by trains of MF stimulations were recorded from OFF-UBCs at a holding voltage of −60 mV (n = 25; *Figure 2*). All OFF-UBCs displayed a slow outward post synaptic current (PSC). This current was blocked by the mGluR2 selective blocker LY341495 (1 μM, n = 25, residual amplitude after block −5.3 ± 10.1 pA; *Figure 2A*). mGluR2 block revealed a NBQX-sensitive inward current in 48% of OFF-UBCs (n = 12) (peak 35.2 ± 28.7 pA; NBQX block of the total charge 95 ± 9%; *Figure 2B*). These AMPA currents did often show a sharp peak at the onset of stimulation, which could result in a net inward current, but the steady-state current was always largely dominated by the outward mGluR2 components. The steady-state amplitude of mGluR2 PSC increased with the stimulation frequency from 10 Hz (41.5 ± 26.9 pA) to 100 Hz (129.9 ± 43.4 pA; n = 14 cells, p<0.001) (*Figures 2C and D*).

mGluR2 currents displayed highly variable activation and decay kinetics (20% to 80% rise time: 259 ± 95 ms; weighted decay time constant 306 ± 149 ms). OFF-UBC fired at the end of the period of low MF activity during 1 Hz modulations and their phase shift was positively correlated with the decay time constant of the mGluR2 current recorded in voltage-clamp (r = 0.621, n = 15, p<0.05) (*Figure 2E*). The total number of spikes per cycle was positively correlated with the amplitude of mGluR2 hyperpolarization (r = 0.627, n = 18, p<0.01), suggesting that rebound-promoting intrinsic

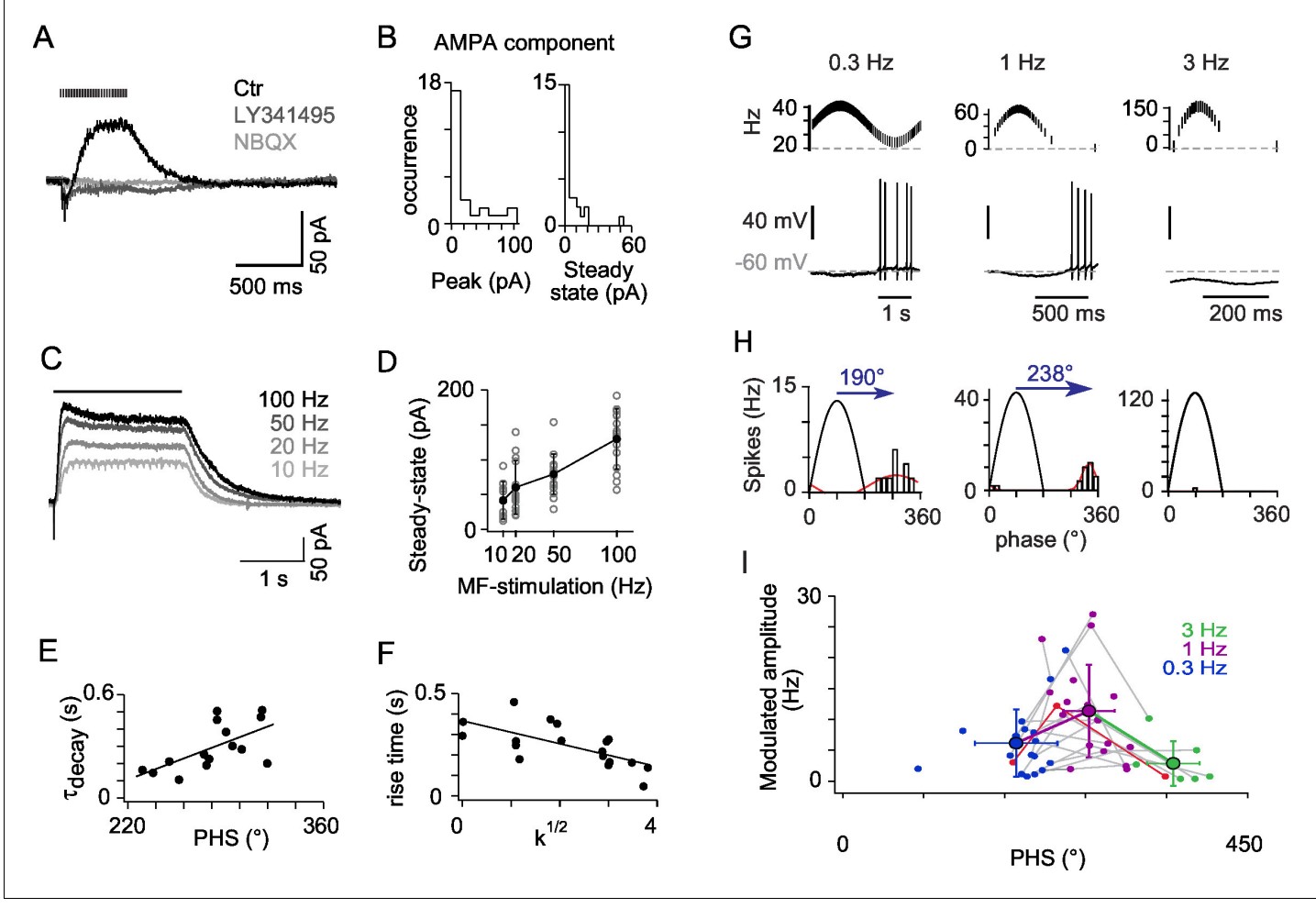

**Figure 2.** mGluR2 synaptic currents govern phase inversion in *OFF-UBCs*. (A) Synaptic currents evoked by a train of MF stimulations (500 ms at 50 Hz) in an *OFF-UBC* in control conditions (ctr; black), after mGluR2 block by LY341495 (0.5 µM, gray), and after block of AMPAR by NBQX (1 µM, light gray). Stimulation train is shown on top. (B) Quantification of the isolated AMPA component in *OFF-UBCs* after mGluR2 block. (C) Steady-state mGluR2 currents evoked by 2s stimulations at 10, 20, 50, and 100 Hz display saturation. (D) Frequency dependence of steady-state mGlur2 synaptic currents. Average is in black. (E) The decay time constant of the EPSC at the offset of 500 ms trains of stimulation at 50 Hz correlates with the phase shift recorded in response to a sinusoidal MF stimulation modulation modulated at 1 Hz rate (*Figure 1*). The black line represents the linear regression. (F) The rise time constant of the outward currents evoked by 500 ms stimulations at 50 Hz correlates with the spikes concentration factor $k^{1/2}$ obtained in *Figure 1*. (G) *OFF-UBC* responses to sinusoidal MF stimulation rate modulations at 0.3, 1, and 3 Hz. Stimulation protocols are shown on top and one period of recording below. (H) Fits of the instantaneous firing rate vs phase relationship of the cell shown in (G). (I) Firing frequency amplitude modulation and phase shifts of 25 *OFF-UBCs* in response to sinusoidal MF stimulation rate modulations at 0.3, 1, and 3 Hz. The cell displayed in (G) and (H) is plotted in red. Average ± SD are indicated for each modulation frequency.

The following source data is available for figure 2:

**Source data 1.** Numerical data corresponding to panels B, D, E, F, I of *Figure 2*.

conductances like $I_H$ or $I_T$ (*Diana et al., 2007*; *Russo et al., 2007*), amplify OFF-UBC responses. The spike burst was interrupted by the rise of the mGluR2 current (see *Figure 2A*), as shown by the negative correlation of the 20 to 80% mGluR2 rise time and concentration factor $k$ (0.668, n = 18, p<0.01).

To characterize the filtering properties of OFF-UBCs, we applied sinusoidal modulations of the MF stimulation rate at frequencies of 0.3 Hz, 1 Hz and 3 Hz to the same cells (*Figure 2G*). Peak and trough firing frequencies of MF stimulations were adjusted to mimic constant amplitude rotations at variable angular frequencies (see Experimental Procedures). As a result, the peak MF firing

frequency increased from ~40 Hz at 0.3 Hz modulation to ~156 Hz at 3 Hz modulation. The amplitude of OFF-UBC responses increased significantly from 0.3 Hz to 1 Hz and dropped at 3 Hz (p=0.0003; Friedman's test; *Figure 2H and I*). Consistently with the increase of PSC amplitude at higher MF firing frequencies (*Figure 2D*), 20% and 68% of the OFF-UBCs were completely silenced during 1 Hz and 3 Hz modulationsrespectively (*Figure 2H and 2I*). Phase shifts increased significantly with the modulation frequency (193 ± 46° at 0.3 Hz; 274 ± 28° at 1 Hz; 367 ± 29° at 3 Hz; p<0.0003; Friedman's test; *Figure 2I*). These results suggest that OFF-UBCs, signal the end of a movement in the non-preferred direction of their input MF by a brief high-frequency burst of spikes.

## ON-UBCs are driven by phasic and buildup AMPA currents

MF EPSCs mediated by AMPARs have previously been recorded from UBCs (*Kinney et al., 1997*; *Rossi et al., 1995*; *van Dorp and De Zeeuw, 2014*). These AMPA EPSCs are characterized by a prominent slow component produced by the entrapment of glutamate in the cleft of the giant MF synapse onto UBCs (*Kinney et al., 1997*; *van Dorp and De Zeeuw, 2014*). These slow EPSCs are likely to mediate the ON behavior and have been proposed to perform some kind of integrative operation on MF activity (*Kinney et al., 1997*; *van Dorp and De Zeeuw, 2014*).

Synaptic currents were evoked by trains of MF stimulations (0.5 s at 50 Hz) to approximate the peak rate and duration of 1 Hz sinusoidal modulation. In all ON-UBCs tested EPSCs were blocked by the competitive AMPAR/kainate antagonist NBQX (1 or 5 µM; block 92.6 ± 8.8% of the charge; n = 10; *Figure 3A*). The time course of the compound AMPA EPSC evoked by MF stimulation trains varied widely between cells (n = 31; *Figures 3A,D*), displaying classical transient currents (*Figure 3B*) as well as slow sustained components. The amplitude of the fast EPSC evoked by the first stimulation ranged from 5 to 210 pA (71 ± 66 pA; n = 21) and was inversely correlated to its decay time course (2.1 ± 1.5 ms) (r = 0.737; p<0.001; n = 21; *Figure 2—source data 1A*). Presynaptic MFs are able to sustain release at 50 Hz without depression (*Saviane and Silver, 2006*). Nevertheless the amplitude of fast EPSCs decreased dramatically (by 67.4 ± 35.8% of the first peak at steady-state, n = 21; *Figure 3B*), in a way which was inversely correlated to the decay time course of the first fast EPSC (r = 0.823; p<0.001; n = 21; *Figure 3C*). Transient EPSCs were always accompanied by a low level of sustained steady AMPAR current (−15.9 ± 8.4 pA before the 4th stimulation; *Figures 3A and B*), indicative of the prolonged presence of glutamate in the synaptic cleft (*Nielsen et al., 2004*).

In 67% of UBCs, an additional slow component developed during the train, a further element suggesting glutamate concentration buildup in the synaptic cleft (*Kinney et al., 1997*). This slow EPSC buildup reached a steady-state (64 ± 35 pA; *Figure 3D*) with variable time course (τ = 0.347 ± 0.244 s at 50 Hz). The amplitude of the transient currents and of the buildup currents were not correlated (p=0.12; *Figure 3E*). The presence of two independent excitatory components suggests that glutamate buildup recruits independent AMPA receptor populations in the two phases (fast and slow) of the synaptic responses. Overall, these data are consistent with the idea that AMPA receptors present at MF to UBC synapses are exposed to glutamate concentration transients of highly variable amplitude and duration which spatio-temporal profiles shape MF EPSCs kinetics through transient activation, steady-state activation and desensitization.

## ON-UBCs EPSCs display a characteristic anomalous rebound at the offset of MF train stimulations

In 87% of all AMPAR expressing UBCs (n = 27 out of 31 cells), an anomalous rebound current developed at the end of the stimulation (*Figure 3A and B*). The 4 remaining cells only displayed slow buildup EPSCs which decayed exponentially with a time constant of 604 ± 361 ms (*Figure 3D*, cell1). Rebound and decay are temporally overlapping processes. The anomalous rebound current was thus isolated by subtraction of the interpolated fitted exponential decay at the offset of the compound EPSC (see Experimental Procedures; *Figure 3F*). The amplitude of the rebound current (27.7 ± 16.7 pA) was correlated to its rise time constant (r = 0.583, p<0.01, n = 23; *Figure 3G*), which varied widely from cell to cell (54 ± 24 ms; min = 25; max = 99; n = 23). The amplitude of the rebound current was also correlated with both the amplitude of the transient EPSC integral (r = 0.611, p<0.01, n = 20; *Figure 3H*) and with the amplitude of the slow buildup EPSC (r = 0.763, p<0.001, n = 25; *Figure 3I*), suggesting that both fast and buildup components of transmission contribute to

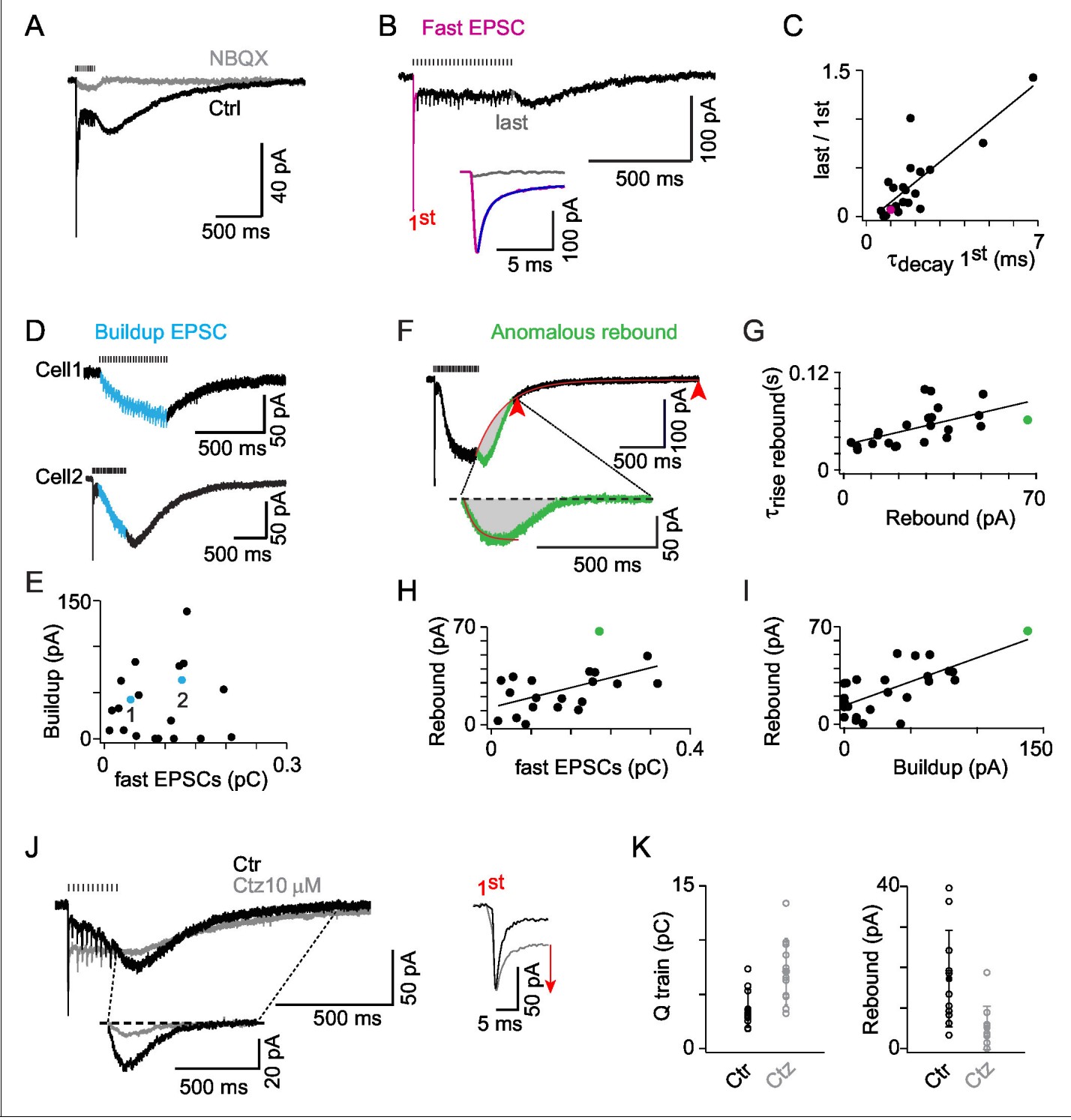

**Figure 3.** AMPAR activation dominates *ON-UBC* synaptic responses and presents three components. (**A**) Synaptic currents evoked by a train of MF stimulations of 500 ms at 50 Hz in an *ON-UBC* before and after perfusion of NBQX (1 μM). Note the transient early EPSC, followed by a slow buildup, and the rebound current at stimulation offset. The stimulation train is shown on top. (**B**) A sustained steady component persists after desensitization of the phasic response during a train of stimuli. The inset shows a superimposition of the fast EPSCs triggered by the first (pink) and the last (gray) stimulations. The exponential fit of the first EPSCs decay is displayed in purple. (**C**) Desensitization of the transient EPSCs at the end of a 500 ms stimulation at 50 Hz correlates to the $\tau_{decay}$ of the transient EPSCs after the first stimulation (r = 0.823; p<0.001; n = 21). (**D**) Examples of slow EPSC buildup in two *ON-UBCs*. (**E**) The slow EPSCs buildup is not correlated to the first transient EPSC amplitude (n = 20). (**F**) Analysis of the anomalous rebound at the end of the stimulation. The last 20% of amplitude decay (red arrowheads) are fitted by an exponential function forced at the last point

*Figure 3 continued on next page*

*Figure 3 continued*

of stimulation (red curve). The isolated rebound current obtained by subtraction of the exponential fit function is displayed in green in the inset. Gray area: charge of rebound current. (G) Correlation of the rebound current amplitude with its rise time (r = 0.583, p =, n = 23). (H) Correlation of the rebound current amplitude with the amplitude of the transient EPSC (r = 0.611, n = 21) (I) Correlation of the rebound current amplitude with the amplitude of the slow EPSCs buildup (r = 0.763, n = 25). Voltage-clamp stimulation protocol in c, d, f, g, h: 500 ms at 50 Hz. (J) AMPA EPSCs before (Ctr, black) and after 10 µM cyclothiazide (Ctz, gray) perfusion. Left bottom inset: isolated rebound current obtained as in (F) in Ctr and Ctz. Right inset: transient EPSCs at the first stimulation in Ctr and Ctz conditions. MF stimulations are shown on top. Voltage-clamp stimulation protocol: 200 ms at 50 Hz. (K) EPSCs charge and rebound amplitude in control condition (Ctr, black) and after 10 µM cyclothiazide perfusion (Ctz, gray; n = 11). Voltage-clamp stimulation protocol in (C), (D), (F), (G), (H), (I): 500 ms at 50 Hz.

The following source data is available for figure 3:

**Source data 1.** Numerical data corresponding to panels C, E, G, H, K of *Figure 3*.

the genesis of the rebound. Cyclothiazide (10–20 µM), a modulator of AMPARs known to suppress desensitization, potentiated the charge during stimulation (201 ± 54% of control; n = 11) and suppressed the rebound current (27 ± 18% of control; n = 11) (*Figures 3J and K*). These results support the idea that rebound is generated by AMPAR recovery from steady-state desensitization (*Kinney et al., 1997*) during slow glutamate clearance from the giant MF synapse onto UBCs (*Billups et al., 2002*; *Rossi et al., 1995*).

## AMPA rebound generates phase dispersion in ON-UBCs

We investigated the link between EPSC time-course and ON-UBC phase shift during sinusoidal MF stimulation. We reasoned that fast and slow synaptic charge occurring during MF trains would favor phase-advanced firing, while rebound charge would favor phase-delayed firing (*Figure 4A,C*). Indeed, the response phase of ON-UBCs at 1 Hz was linearly correlated with the ratio between the rebound charge and the synaptic charge during the 50 Hz trains (see Experimental Procedures) (slope = 0.1 per 100°; r = 0.734, p<0.001, n = 22; *Figure 4D*). Neither the charge during the train alone (p>0.1; r = 0.213; n = 16; *Figure 2—source data 1B*), nor the onset time course of the buildup component did correlate with the phase (p>0.1; r = 0.036; n = 14; *Figure 2—source data 1C*).

In all delayed cells (positive phase shift), the phase of UBC firing was strongly correlated with the charge of the rebound EPSC (r = 0.75, p<0.001, n = 16; *Figure 2—source data 1D*). A mechanistic explanation for this result could be that larger rebounds have slower time to peak (*Figure 3g*). Indeed we found that the onset time constant of the rebound was strongly correlated to the phase of UBC firing (r = 0.87, p<0.001, n= 16). When phase was converted to a time delay to maximal firing (2.78 ms deg$^{-1}$) the regression slope was close to 0.33 (r = 0.87, p<0.001, n= 16 *Figure 4E*) indicating that maximum firing occurs near the peak of the rebound current (3 onset time constants). Cyclothiazide (*Figure 4F*), at a concentration which suppresses the rebound current, shifted the phase of UBC firing (143 ± 44° in control; −59 ± 36° in cyclothiazide; n = 7; p=0.01; *Figure 4G*) towards values similar to control cells displaying small rebound (p=0.9, n = 6 cells with little rebound current, n = 7 cells in cyclothiazide). Hence anomalous rebound mediates phase inversion in the response to modulated MF inputs.

The frequency dependence of the UBC response was then tested by implementing sinusoidal modulations of the MF stimulation rate at frequencies of 0.3 Hz, 1 Hz and 3 Hz (*Figure 4H*). In contrast to OFF-UBCs the average amplitude of firing modulation increased at higher MF modulation frequencies (5.1 ± 5.9 Hz at 0.3 Hz, 11.0 ± 8.1 Hz at 1 Hz, and 14.0 ± 9.7 Hz at 3 Hz; p=0.0009; Friedman's test). Interestingly, the frequency of maximal modulation varied from cell to cell. The population response of UBCs also showed significant frequency-dependent increase of phase lag (45.9° at 0.3 Hz, 108.9° at 1 Hz and 191.6° at 3 Hz; p<0.0001; Friedman's test). and remained broadly distributed at all frequencies (SD: 76.3°, 73.9° and 92.7° at 0.3 Hz, 1 Hz and 3 Hz respectively; *Figure 4I*). Phase dispersion of the ON-UBC responses is therefore a property preserved over a wide range of movement dynamics.

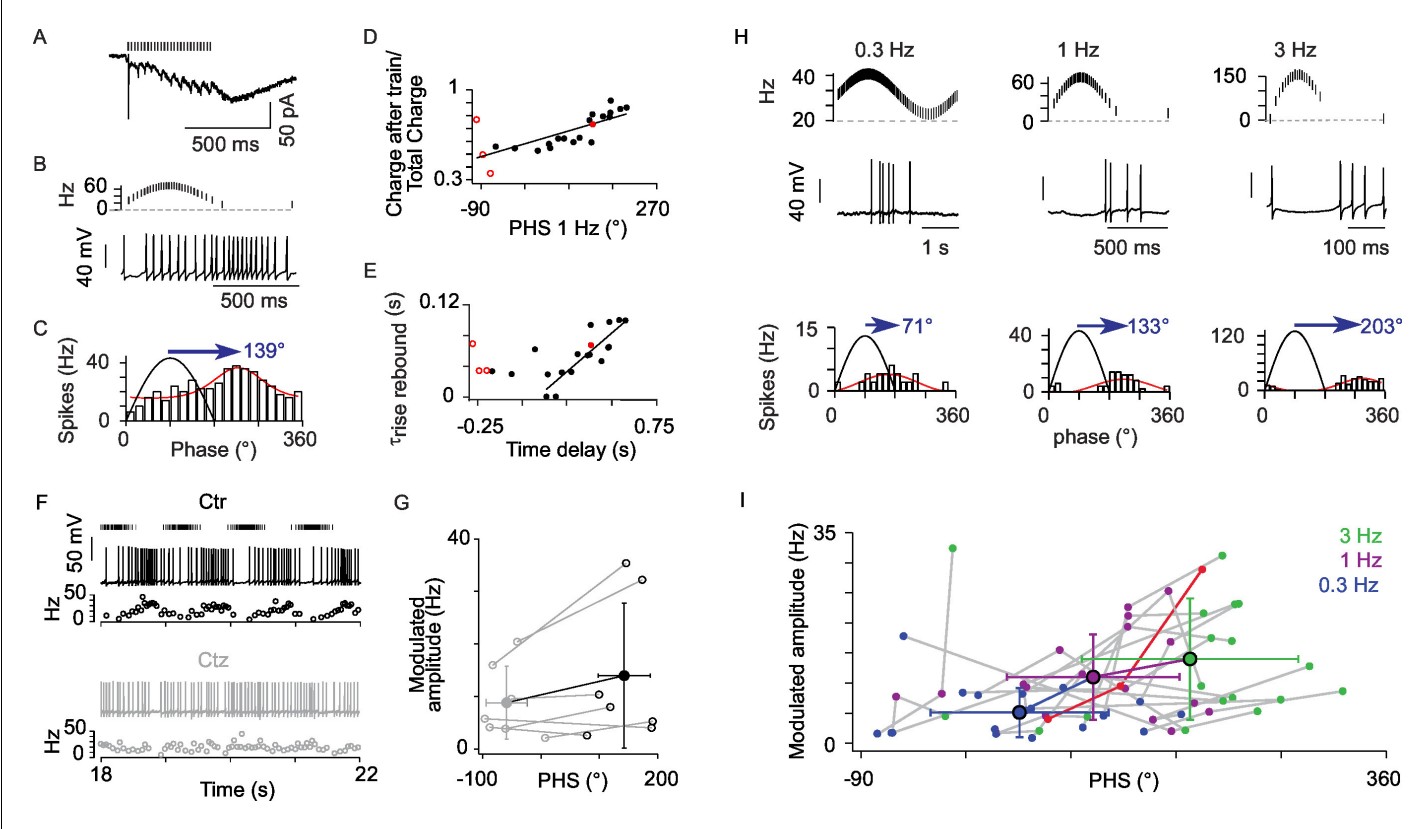

**Figure 4.** The AMPA rebound current defines the phase shift of UBC responses to sinusoidal input modulations. (**A**) AMPA EPSCs of an ON-UBC with early phase shift. (**B**) Voltage response of the same UBC shown in A. (**C**) Circular Gaussian fit of the firing rate distributions of the cell shown in (**B**). (**D**) Correlation of the response phase at 1 Hz to the ratio charge at the offset of stimulation / total synaptic charge (r = 0.734, p<0.001, n = 22). (**E**) Correlation of the time delay to maximal firing to $\tau_{rise}$ of the rebound current (r = 0.87, p<0.001, n= 16). (**F**) Example of an ON-UBC's voltage response to 1 Hz sinusoidal modulations of MF stimulation (4 cycles shown out of 10) before (Ctr) and after 10 μM cyclothiazide (Ctz) perfusion. Stimulations are shown on top. (**G**) Phase shift in control condition (black) and in 10 μM cyclothiazide perfusion (gray, n = 7). (**H**) ON-UBC responses to sinusoidal MF stimulation rate modulations at 0.3, 1, and 3 Hz. Stimulation protocols are shown on top and one period of recording below. Circular Gaussian fits of the firing rate distributions of the same cell. (**I**) Modulated firing amplitudes and phase shifts of 25 ON-UBCs in response to sinusoidal MF stimulation rate modulations at 0.3, 1, and 3 Hz. The cell displayed in (**H**) is plotted in red. Average ± SD are indicated for each modulation frequency. Voltage-clamp stimulation protocol in (**D**), (**E**): 500 ms at 50 Hz.

The following source data is available for figure 4:

**Source data 1.** Numerical data corresponding to panels D, E, G, I of *Figure 4*.

## Diverse synaptic glutamate time course shapes synaptic currents and explains phase dispersion in ON UBCs

Modeling of glutamate concentration profiles in the MF to UBC synaptic cleft has shown that the average frequency dependent behavior of the peak and steady-state conductance can be accounted for by a detailed Markov model of the AMPA receptor (*van Dorp and De Zeeuw, 2014*). To further investigate the role of synaptic glutamate profiles in the cell to cell variability of UBC responses we built a simplified model with reduced parameter space. First, we designed the minimal Markov scheme which can account for a bell-shaped steady-state concentration-response curve with non-zero saturating steady-state current, as recorded in UBCs (*Kinney et al., 1997*). This model includes one close state, two open states and one desensitized state (*Figure 5A*) (See Experimental Procedures). The on-rate of desensitization was adjusted to account for the depression of the fast receptor current during MF trains and the recovery rate was adjusted to yield a steady-state activation curve with a peak of 16% of the total conductance at 25 μM (*Figure 5B*) (Experimental Procedures) and a

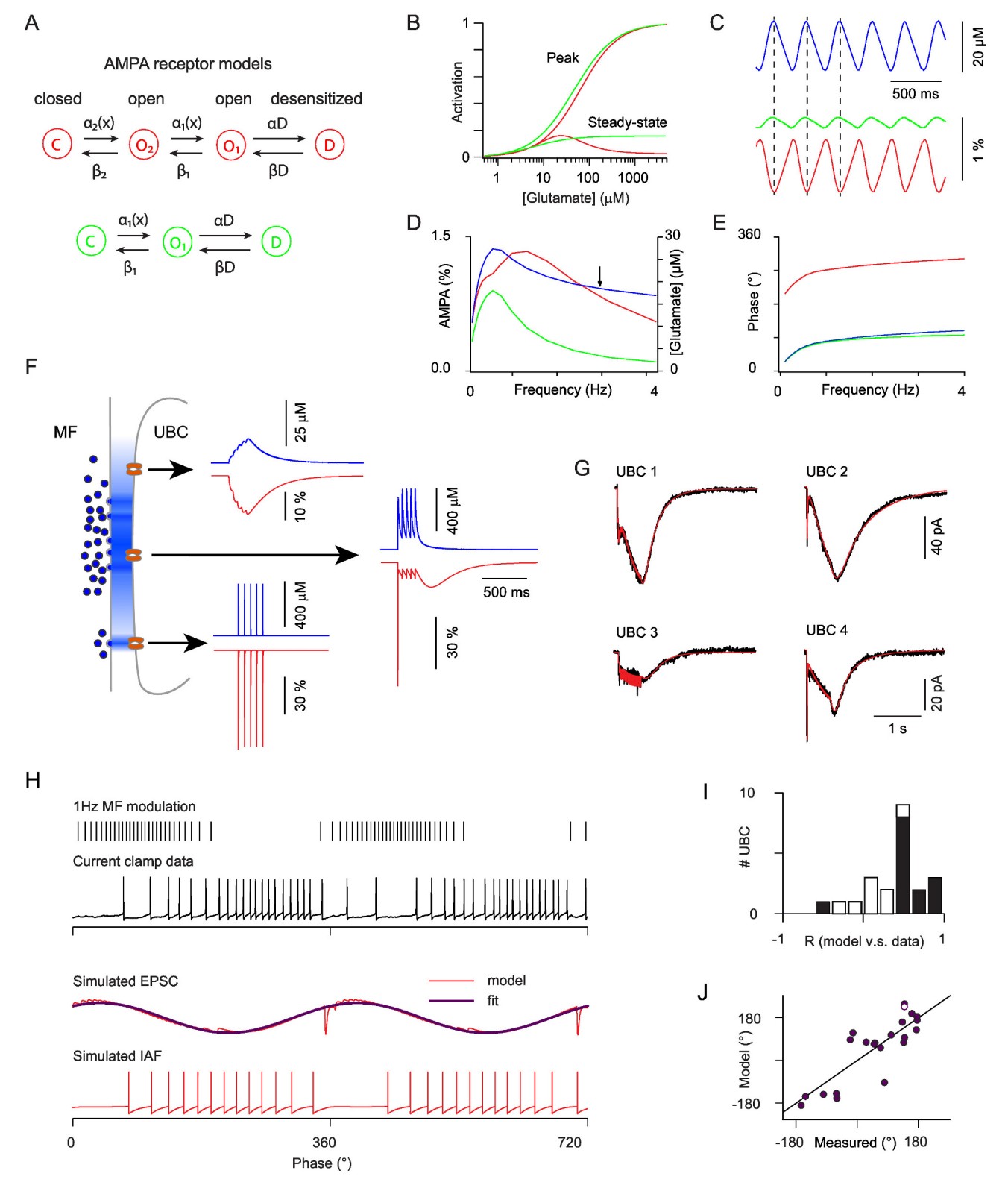

**Figure 5.** Modeling the AMPAR responses of *ON-UBCs*. (**A**) Simple Markov models for desensitizing AMPARs with bell-shaped steady-state activation curve (red) and monotonous steady-state activation (green). (**B**) Peak and steady-state dose-response curves for the red and green models in (**A**). (**C**) Simulated glutamate concentration (blue) and AMPAR activation (red and green) during a 3 Hz sinusoidal stimulation of the presynaptic mossy fiber (same parameters as for the experimental data). Each presynaptic action potential was set to produce a glutamate transient of 2 µM which rose in 15

*Figure 5 continued*

ms and decayed exponentially with a t of 600 ms. (D) Peak to peak modulation amplitude of the glutamate concentration and fractional AMPAR activation (colors as in C) as a function of the MF firing rate modulation frequency. (E) Phase of the glutamate concentration and AMPAR response relative to the MF modulation. Note the phase inversion of the bell-shaped model. (F) Schematic depiction of glutamate concentration profiles (in blue) at the giant MF-UBC synapse upon multivesicular release. Simulated responses (glutamate in blue, bell-shaped AMPAR Markov model in red) to a train of 5 stimulation at 20 Hz are shown for the following parameters (peak, $\tau_{rise}$, $\tau_{decay}$) of action-potential evoked glutamate transients: top inset (4 µM, 20 ms, 250 ms), middle inset (500 µM, 1 ms, 250 ms), lower inset (500 µM, 0 ms, 1.5 ms). (G) Example of the response of 4 *ON-UBCs* to a 500 ms train of MF stimulation at 50 Hz and of their fit (red curve) by the summed activation of three AMPAR populations submitted to different glutamate transients, as in (F). (H) Response of an *ON-UBC* to a sinusoidal modulation of its afferent MF firing rate at 1 Hz (black trace) and simulated EPSC obtained with the parameters obtained through fitting in (G) (middle red trace). The response of a simulated integrate and fire neuron to this EPSC is shown in the bottom trace. Purple trace, exponentiated cosine fit of the EPSC used in (J). (I) Correlation between the simulated phase-dependent EPSC and the experimental phase-dependent firing response of UBCs (n = 22). Significant correlations (p=0.05) are displayed in black bars. (J) Correlation between the phase shift of the simulated EPSC and the measured phase shift of the UBC response in current clamp. Line represents equality. Hollow circle: cell displayed in (H).

fractional steady state conductance at saturating glutamate concentration of 2.5%. This simple Markov model was compared to a three state model with similar affinity and maximal steady-state current, but displaying a monotonic steady-state concentration response curve (*Figures 5A and B*). When submitted to slow synaptic glutamate buildup (*Figure 5C*) at increasing firing modulation frequencies, for which stable glutamate modulation amplitudes are reached (*Figure 5D*, blue), the three state model saturated above 0.7 Hz due to basal glutamate buildup (*Figure 5D*, green) while the four state model showed a resonance above 1 Hz and greatly increased modulation bandwidth (*Figure 5D*, red). This resonance was directly linked to the steady-state rebound, as attested by a 180° phase shift of the response compared to glutamate, as opposed to the three state model situation (*Figure 5E*). Hence, the bell-shaped steady-state response of the AMPARs at UBC synapses plays a central role to encode the time integral of modulated inputs over a wide frequency bandwidth.

To explain the phase diversity of UBCs we reasoned that AMPARs at distinct locations in the giant MF to UBC synaptic articulation will face diverse synaptic glutamate transients, thus imparting varied response dynamics. If located at the center of the synaptic area AMPARs will sense a large and prolonged glutamate transient that will build up during the train (*Figure 5F*, middle trace). Receptors located at the periphery of the synaptic cleft will either sense low amplitude and slow glutamate buildup (*Figure 5F*, upper trace) or a fast glutamate transient if they are located in front of a synaptic release site (*Figure 5F*, lower trace) (see Experimental Procedures for detailed model descriptions). We were able to fit accurately the complex time course of synaptic currents in all UBCs (see a few examples in *Figure 5G*). The SD of the residual error was on average 2.55 ± 0.28 pA (mean ± SEM; median = 2.25), which was much smaller than the amplitude of the currents (18.82 ± 4.25 pA; median = 13.54), and of the same order as the fluctuations in the current traces as measured in the 50 ms before stimulation (0.71 ± 0.08 pA; median = 0.63). Thus, while AMPAR properties explain the presence of an anomalous rebound, the time course of synaptic glutamate transients can fully account for the diversity of synaptic responses at MF to UBC synapses.

We then used the parameters obtained by fitting the EPSCs to compute the expected AMPA response to sinusoidal MF modulations (*Figure 5H*). In most cells (14 out of 22) phase histograms of the simulated EPSCs and of the experimentally recorded UBC firing rate were significantly correlated (p<0.05; t-test; *Figure 5I*). Fitting the simulated EPSC by a sinusoidal function yielded a simulated phase shift. Simulated and experimental phase shifts were linearly correlated (R = 0.89 and p=2e-5; t-test and slope = 1.04 for cells with correlated phase histograms) (*Figure 5J*). In the cells displaying significant correlation between the EPSCs and firing rate, the phase difference between modeled and experimental data was approximated by a Gaussian distribution (−24.6 ± 64.3) and experimental firing could be approximately reproduced using a simple integrate and fire model (*Figure 5H*). These data confirm that the dynamics of glutamate accumulation at the MF to UBC synapse during time-modulated MF firing is the major determinant of firing phase in UBCs.

## UBCs increase the phase diversity of GCs responses

We next investigated the computational consequences of the diversity of phase shifts in the UBCs responses. We thus simulated a granular layer network with 4500 Granule Cells (GCs) receiving inputs from 500 MFs, that were either extrinsic (eMFs) or from UBCs. The network structure is illustrated in *Figure 6A*. We compared the response of GCs to sinusoidal eMF stimulation, in the presence and absence of UBCs. In the model network, each GC receives 4 inputs (*Palay and Chan-Palay, 1974*). In the network without UBCs, all 4 inputs are eMFs (*Figure 6A*, left). In the network with UBCs, each input is randomly and independently set (*Figure 6A*, right) to be from either an eMF or a UBC with equal probability (0.5).

We simulated the dynamics of the granular layer model in response to sinusoidal vestibular inputs. We used the same stimulation protocols as in our experiments, with three different frequencies (*Figure 6B*, blue). All eMFs inputs were either in phase or in anti-phase with modeled sinusoidal movements (*Arenz et al., 2008*). Parameters of model UBCs were chosen by randomly sampling parameters fitted from recorded UBCs (*Figure 6B*, red). This led to a large diversity of phase shifts. As a consequence, a subset of UBCs fired at any phase of the oscillatory cycle. Spike trains from eMFs and iMFs were then injected into the GCs (see Experimental Procedures for stimulation details). The mean conductance from eMFs/UBCs to GCs was adjusted to match the EPSC amplitudes reported in the literature (*Arenz et al., 2008*; *Schwartz et al., 2012*). To introduce heterogeneity in the network, all synaptic conductances were drawn randomly from a Gaussian distribution (see Experimental Procedures), whereas, for the sake of simplicity, the inhibition from Golgi cells was replaced by a constant inhibitory conductance (*Billings et al., 2014*) (see Discussion). The responses of simulated GCs are shown in *Figure 6C*. Without UBCs, GCs fire in phase with eMFs inputs, as shown in *Figure 6Ci*. The spike trains of 30 randomly selected GCs in a single trial with 1 Hz stimulation and their average firing rate over trials are shown in *Figure 6Ci* (left). The amplitude of modulation and phase shift from the fitted GC firing rate curves were plotted in a polar plot (*Figure 6Ci* (right)). We found that the preferred firing phases were in phase with the extrinsic vestibular mossy fibers, i.e. in phase or antiphase with simulated movement. When UBCs are included in the network, the response of GCs shows much more heterogeneous phase shifts. The distribution of phases in the network comprising UBCs is significantly closer to a uniform distribution compared to the network without UBCs (KS distance between network with UBCs and uniform: 0.1673; between network without UBCs and uniform: 0.4134; both distributions are significantly different, KS test, p<1e-10 (*Figure 6Cii*), similar to experimental in vivo data (*Barmack and Yakhnitsa, 2008*). Therefore, the heterogeneity of UBCs phase shifts leads to a diversity of GCs phase shifts.

## Purkinje cell learning of arbitrary input-output relationships depends on the presence of UBCs

We then investigated the functional consequences of this GC phase shift variability on downstream Purkinje cells (PCs). In particular, we asked to which degree the heterogeneity of GC phase relationships has an impact on the ability of a PC to learn arbitrary input-output relationships. We consider a single PC that receives inputs from all simulated GCs of the network. The parallel fiber to Purkinje cells synapses are endowed with a standard climber fiber (CF)-dependent learning rule (see Experimental Procedures), such that synapses are depressed upon coincidence of presynaptic GC and CF activity, while they are potentiated when presynaptic GCs are active during CF silence (*Dean et al., 2010*). We tested the ability of the PC to learn to fire preferentially at arbitrary phases of the sinusoidal eMF drive, by implementing an "error signal" in the CFs, proportional to the difference between actual and desired PC outputs. *Figure 7* shows that in a network with UBCs, the PC can learn to fire at arbitrary target phases, even though eMF inputs peak only at two possible phases (*Figure 7A*, top and *Figure 7B*). This ability breaks down in a network without UBCs, in which PCs can only learn to fire in phase with eMF inputs (*Figure 7A*, bottom and *Figure 7B*). This is consistent with experiments showing that the response of PCs, similarly to GCs, has a large degree of heterogeneity (*Barmack and Yakhnitsa, 2008*).

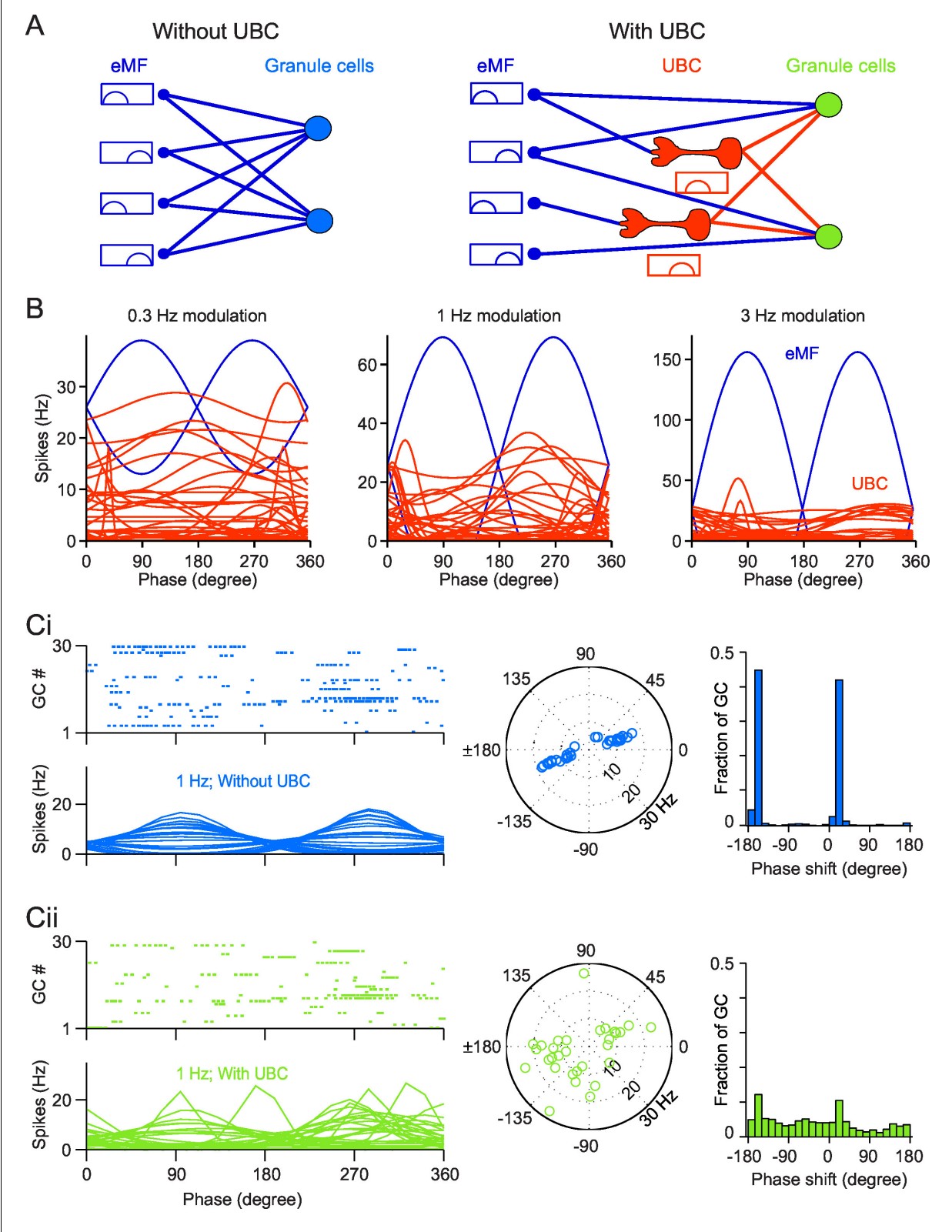

**Figure 6.** Simulation of a Granular network with/without UBCs. (**A**) Illustration of the Granular network. Granular cells receive inputs from only eMFs without UBCs (left) and both eMFs and UBCs (right),. UBCs have different phase shifts. (**B**) Firing rate vs phase curves of simulated eMFs (blue) and all recorded UBCs (red) at 0.3 Hz (left), 1 Hz (middle) and 3 Hz (right) modulations. (**Ci**) Sample Granular cell outputs without UBCs in the network. (left,

*Figure 6 continued on next page*

*Figure 6 continued*

upper) Raster plots of 30 random selected GCs in one input period, and (left, bottom) their averaged firing rate curve; (right) polar plots of GC phase shifts. (**Cii**) Same as (**Ci**), but for sample Granular cell outputs with UBCs in the network.

## Discussion

We have provided here a detailed characterization of the filtering properties of UBCs in the presence of sinusoidal MF inputs. We have shown that UBCs encode these inputs with a surprising degree of heterogeneity, their responses spanning all the possible range of phase shifts, at all the frequencies tested here (from 0.3 to 3 Hz). Our study reveals that populations of UBCs in the vestibular cerebellum behave as a bank of diverse temporal filters. By virtue of this temporal diversity, it is possible to find GCs that fire at any phase of a sinusoidal head movement, as hypothesized by adaptive filter models (*Dean et al., 2010*; *Fujita, 1982*; *Gao et al., 2012*; *Roberts and Bell, 2000*), and consistent with in vivo data (*Barmack and Yakhnitsa, 2008*). The UBC therefore extends the combinatorial notions of pattern separation (*Marr, 1969*) and expansion recoding (*Albus, 1971*) into the temporal domain. In turn, this diversity of temporal representations allows PCs to learn arbitrary phase response functions, as shown here by implementing a classical climbing fiber-driven learning rule that modifies the weights of parallel fiber to PC synapses. Our results are consistent with a recent study in electric fish, which showed that delayed responses necessary for predicting sensory consequences of motor commands are generated by UBCs in the eminentia granularis posterior (EGp) of the mormyrid electric fish, a structure similar to the granular layer of the mammalian

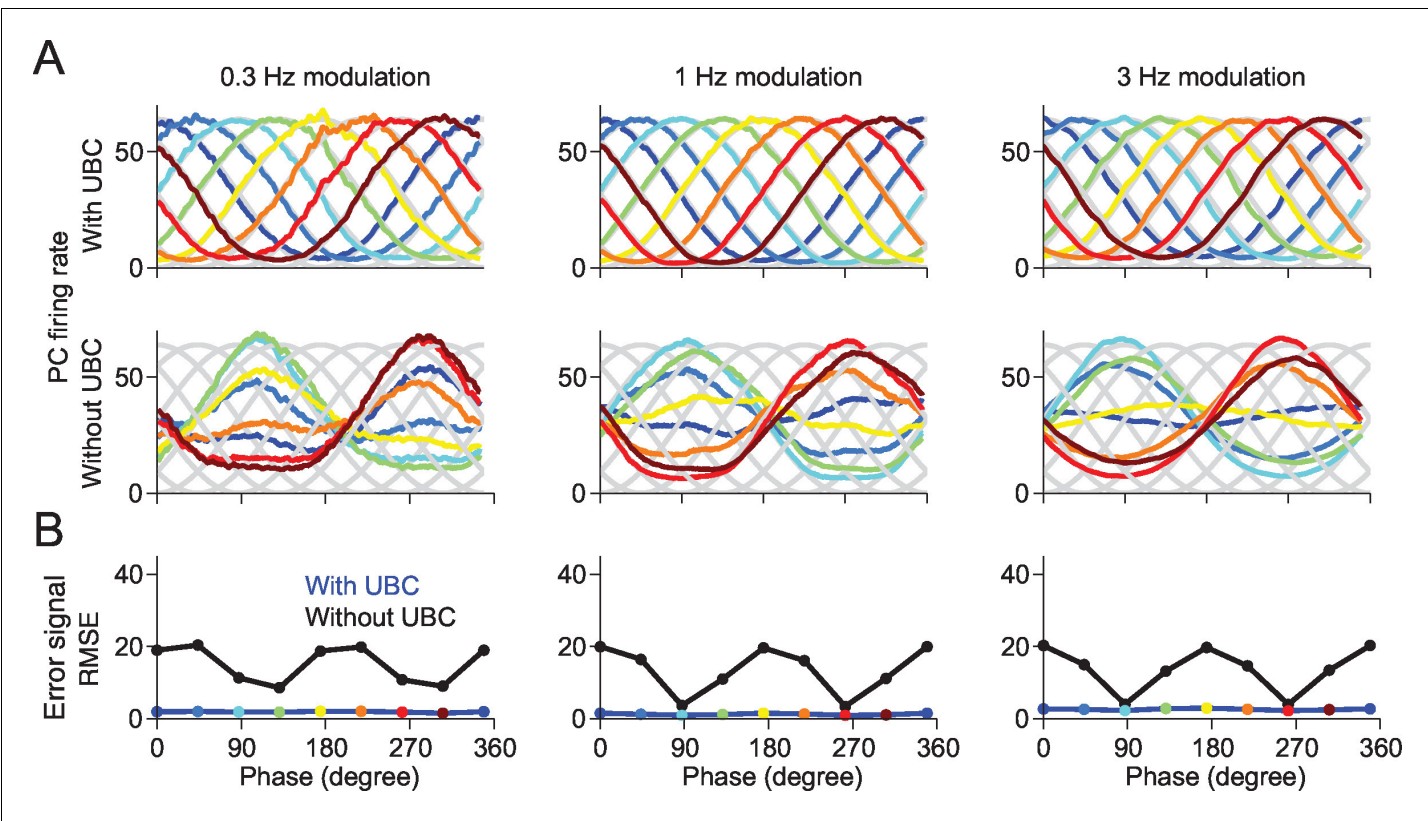

**Figure 7.** A downstream Purkinje cell can learn to fire at arbitrary phases in a network containing UBCs. (**A**) The responses of the Purkinje cell are tuned to the given target phase in the network with UBCs. Each target phase is indicated by each color. (**B**) The mean square error (MSE) between the target and the response of Purkinje cell is systematically smaller in the network with UBC (blue) than those without UBC (black). Each color point indicates the PC's response shown in (**A**).

cerebellum (*Kennedy et al., 2014*). An interesting question for future research is whether the properties of UBCs in mammals are set to optimize performance of the cerebellar circuit, given the natural statistics of vestibular inputs. The differential distribution of calretinin-expressing (type I) and mGluR1-expressing (Type II) UBCs in vestibular lobules and parasagittal band within these lobules (*Consalez and Hawkes, 2012*; *Mugnaini et al., 2011*) support the idea of some functional specialization. Additionally, the presence of UBCs in the dorsal vermis and cerebellar hemispheres of higher mammals (*Dino et al., 1999*) suggests that the temporal filter properties described in this work may also be useful outside the context of vestibular sensory processing.

We have established that the diversity of UBC responses is due to a combination of two factors: (1) the heterogeneous expression of ionotropic and metabotropic glutamate receptors (see also *Borges-Merjane and Trussell, 2015*) and (2) the variability of AMPAR-mediated responses, due to cell-specific combinations of transient, build-up and rebound synaptic components. The heterogeneity of UBCs has already been recognized (*Mugnaini et al., 2011*), particularly in relation to the expression of metabotropic glutamate receptors (*Borges-Merjane and Trussell, 2015*; *Kim et al., 2012*; *Nunzi et al., 2002*; *Rousseau et al., 2012*; *Russo et al., 2008*). However, their contribution to MF synaptic transmission for dynamic inputs was unknown. The paucity of ionotropic receptors at some mGluR2-expressing MF-UBC synapses is also surprising, given the size of the synaptic articulation and the number of synaptic release sites (*Mugnaini et al., 1994*). As a consequence, cells expressing mGluR2 receptors (OFF UBCs) fire in antiphase, or in advance of the inputs. Adding further complexity to this temporal diversity, our data (*Figure 1—source data 1*) suggest that the relevant subpopulation of mGluR1-expressing UBCs (*Borges-Merjane and Trussell, 2015*; *Mugnaini et al., 2011*), not studied here, participate to the very slow integration of the MF inputs. The predominance of mGluR transmission at the MF synapse is not directly linked to the giant synaptic articulation, as mGluRs are exclusively found at extrasynaptic sites on the dendritic appendages of the UBC brush (*Jaarsma et al., 1998*). mGluR activation, which occurs at standard synaptic articulations on spines or dendritic shafts upon repetitive stimulations, may thus implement temporal integration of population activity in many other neuronal types throughout the CNS.

Cells expressing AMPARs (ON UBCs) fire with a large diversity of phase shifts. We show that the diversity of phase shifts in AMPAR-mediated responses could be well described by exposing AMPARs to various types of synaptic glutamate transients. To account for late-phase responses and for the wide bandwidth of UBC modulations, the AMPAR must present a bell-shaped steady-state response curve, as described experimentally (*Kinney et al., 1997*). The combination of these properties and of spillover-induced desensitization at the UBC synapse generates strong rebound currents following a train of MF action potential (see also *van Dorp and de Zeeuw, 2014*). These properties differ dramatically from the fast reliable transmission observed at other giant synapses like the calyx of Held, where low release probability and fenestration of the apposition limit glutamate spillover and AMPAR desensitization (*Taschenberger et al., 2002*). We hypothesize that a combination of factors, including the complex geometry of MF-UBCs synapse, the variety of AMPARs and auxiliary subunits (*Jackson and Nicoll, 2011*), and the localization of AMPARs with respect to release sites (*Jaarsma et al., 1995*), could explain the variety of AMPA EPSCs profiles at MF to UBC synapses. Our results strongly support the importance of the desensitized states for the functional properties of ionotropic glutamate receptors.

Other sources of differential phase responses in the vestibular system have been described in the literature. Primary sensory cells have been shown to display different levels of non-linearity in response to sinusoidal movements which appear related to cellular excitability (*Pfanzelt et al., 2008*). In the vestibular nuclei, diversity of single cell properties (*Sekirnjak and du Lac, 2002*) and synaptic dynamics of primary afferences (*Broussard, 2009*) could in principle generate a diversity of secondary mossy fiber inputs to the vestibulo-cerebellum, though the range of phase shifts shown in those reports is fairly modest and much smaller than the one shown here. Indeed, *Arenz et al. (2008)* showed highly stereotyped rate modulation of mossy fiber inputs to vestibulo-cerebellar GCs during head rotation with little phase shift. In GCs, a diversity of afferent mossy fibers synaptic properties could lead to a diversity of delays (*Chabrol et al., 2015*) in the range of tens of milliseconds, much smaller than needed to cover the phase spectrum during head movements. In the cerebellar models of delayed eyelid conditioning (*Kalmbach et al., 2010*), intrinsic dynamics of the granular layer circuit involving feed-forward and feed-back inhibitory interactions have been proposed to produce delay-specific granule cell activity patterns (*Kalmbach et al., 2011*; *Medina et al., 2000*;

Rössert et al., 2015; Yamazaki and Tanaka, 2007). These other sources of diversity in phase responses might allow the cerebellar circuitry to learn input-output associations with significant delays even in circuits that lack UBCs. However, we showed here that UBCs can generate much larger delays, as well as OFF-responses, within a wide range of stimulation frequencies. The variable tuning frequency of UBCs could explain the frequency-specific learning of the VOR gain modulation (Lisberger et al., 1983), because different populations of UBCs and hence of granule cells would be optimally modulated for different rotation frequencies. Finally UBCs have the capability to generate high-frequency rebound bursts (Diana et al., 2007; Locatelli et al., 2013), as shown here for some of the OFF-UBCs, which may provide useful timing information, as suggested in the electro-sensory lobe of the electric fish (Kennedy et al., 2014; Sawtell, 2010). Golgi cell inhibitory inputs to UBCs (Dugue et al., 2005), which were pharmacologically blocked in this study, may enhance this rebound behavior (Kennedy et al., 2014), providing for richer interactions between the granular layer dynamics and UBC intrinsic excitability. Our data support a preferential role of UBCs in tasks in which non trivial associations between inputs and outputs at different phases need to be learned, like phase-shifted VOR or high-velocity compensatory eye movements during which non-linear inverse eye dynamics have to be computed (Green et al., 2007; Lisberger, 2009). Optogenetic suppression of vestibular UBC activity should lead to significant impairment in these motor tasks. Furthermore the higher prevalence of UBCs in the hemispheric lobules in higher mammals (Dino et al., 1999) argues for the involvement of this cell-type in complex motor and cognitive tasks involving the cerebellum.

Our results fit well with a growing number of studies that have emphasized the benefits of heterogeneity for coding in neural circuits. Shamir and Sompolinsky, (2006) found that neuronal heterogeneities allow networks to overcome the drastic limits imposed by neuronal correlations on the accuracy of population codes (see also Ecker et al., 2011). The benefits of neuronal heterogeneities have also been pointed out in several specific sensory systems. Padmanabhan and Urban, (2010) showed that the diversity of intrinsic properties of mitral cells of the olfactory bulb allows a population of such cells to double the information contained in their responses, compared with a homogeneous population. Gjorgjieva et al., (2014) examined the benefits of 'pathway splitting' in sensory systems, focusing on the emergence of ON and OFF pathways in the retina. A recent further showed the benefit of two types of OFF-cell thresholds for maximizing information transmission (Kastner et al., 2015). Most of these studies focused on heterogeneities in static single neuron properties. Here, we have shown that the benefits of heterogeneities extend into the temporal domain, and can be exploited by a read-out that can learn arbitrary temporal input-output relationships.

## Materials and methods

### Slice preparation

All experiments were performed according to the ethics rules of the Centre National de la Recherche Scientifique and protocols were approved under number 02235.02 of the general agreement C750520. Wistar rats (22 to 30 days old) were deeply anesthetized with isoflurane and the vermal part of the cerebellum was isolated in a cold bicarbonate based solution (BBS) containing (in mM): 125 NaCl, 3.5 KCl, 1.25 $NaH_2PO_4$, 25 $NaHCO_3$, 20 glucose, 1.6 $CaCl_2$, 1 $MgCl_2$, and 0.00005 minocycline, (oxygenated with 95% $O_2$, 5% $CO_2$). Parasagittal slices (290 mm) were immediately obtained with a vibratome (HM 650 V; Microm) while preparation was kept in an ice-cold cutting solution composed of (in mM): 130 K-gluconate, 15 KCl, 2 EGTA, 20 HEPES, 25 glucose, 0.05 D-(-)-2-Amino-5-phosphonovaleric acid (D-AP5; Tocris), and 0.00005 minocycline (pH adjusted to 7.4 with NaOH). Slices recovered initially at 33°C for 30 s in a modified BBS solution (in mM): 225 D-mannitol, 2.34 KCl, 1.25 NaH2PO4, 25 NaHCO3, 25 Glucose, 0.513 CaCl2, 7.671 MgCl2, 0.05 D-AP5, and 0.00005 minocycline (oxygenated with 95% $O_2$, 5% $CO_2$). Then slices were incubated in oxygenated BBS at 33°C for up to 6 hr.

### Electrophysiology

Slices were placed in a recording chamber mounted on an upright microscope (BX51W, Olympus), visualized with deep red light (750 ± 25 nm) and a CoolSnap SF CCD camera (Photometrics). The preparation was elevated on a nylon grid to promote continuous perfusion of a bubbled BBS

solution (3 ml/min) under the slice. Electrophysiological recordings were obtained at 34–36°C from UBCs in the lobule X of the cerebellar vermis. Voltage- and current-clamp recordings were performed in the whole-cell configuration. Data were sampled at 40–50 kHz and filtered at 5 kHz acquired using an EPC10 amplifier and the Patchmaster software (HEKA). Patch pipettes (3.5–4 MΩ) were pulled from borosilicate glass capillaries (Hilgenberg) with a vertical puller and filled with an intracellular solution containing (in mM): 135 KMeSO$_4$, 3 NaCl, 1MgCl$_2$, 0.1 EGTA, 10 phosphocreatine-K$_2$, 10 HEPES, 4 ATP-Mg, and 0.4 GTP-Na$_2$ (pH adjusted to 7.35 with KOH; osmolarity adjusted to 295 mOsm). EGTA was replaced with 10 mM BAPTA and KMeSO$_4$ concentration was decreased to 114 mM in a first set of experiments (n=25) aimed at recording the synaptic currents only. Alexa Fluor-488 (20 µM; Invitrogen) was added to the pipette solution to visualize the cell morphology and easily identify UBCs. Cell membrane capacitance ranged from 7 to 20 pF. Step current injections (5 to 50 pA) from a membrane potential of −80 mV were used to test the capability of UBCs to generate high frequency bursting activity. Pipette for mossy fiber stimulation was filled with HEPES-buffered solution and it was placed in the granule cell layer at more than 100 µm from the cell body. The stimulation protocols were obtained by modeling eMF activity as $r_{eMF}[1 + Asin(2\pi ft)]_+$ with $r_{eMF}$=26 Hz, and the modulation $A = \frac{5}{3}f$ is obtained from in vivo data at frequency $f = 0.3$ (**Arenz et al., 2008**). The stimulation consists in a constant fire rate 26 Hz for 10 s, and then a modulated rate for another 10 s at three different frequencies 0.3, 1 and 3 Hz, respectively. GABAergic and glycinergic components were blocked during the experiments by perfusion of 6-imino-3-(4-methoxyphenyl)-1(6H)-pyridazinebutanoic acid hydrobromide (SR95531; 2 µM; Abcam Biochemicals), (2S)-3-[[(1S)-1-(3,4-dichlorophenyl)ethyl] amino-2-hydroxypropyl] (phenylmethyl) phosphinic acid hydrochloride (CGP 55845; 0.5 mM; Tocris), and strychnine (1 µM). Synaptic currents were identified by bath application of antagonists. mGluR1 component was blocked with (3,4-Dihydro-2H-pyrano[2,3-b]quinolin-7-yl)-(cis-4-methoxycyclohexyl)-methanone (JNJ 16259685; 1 mM; Tocris). mGluR2 component was blocked with (2S)-2-amino-2-[(1S,2S)-2-carboxycycloprop-1-yl]-3-(xanth-9-yl) propanoic acid (LY 341495; 0.5 µM; Tocris). AMPA component was blocked with 2,3-Dioxo-6-nitro-1,2,3,4-tetrahydr obenzo[f]quinoxaline-7-sulfonamide disodium salt (NBQX; 1 mM; Abcam Biochemicals). AMPA receptor desensitization was modified by the allosteric modulator 6-Chloro-3,4-dihydro-3-(5-norbornen-2-yl)-2H-1,2,4-benzothiazidiazine-7-sulfonamide-1,1-dioxide (cyclothiazide; Abcam Biochemicals) at the specified concentrations.

## Data analysis

Data were analyzed with Igor PRO (WaveMetrics) and spikes detection was performed with the custom-made threshold-detection algorithm SpAcAn (**Dugué et al., 2005**; http://www.spacan.net/). Duration of the membrane potential depolarization after the end of sinusoidal MF modulation in complex cells was calculated as the time between the last stimulation and the time at which the membrane potential decays at the resting membrane potential value. The phase shift of UBC firing rate modulation was obtained by fitting single spikes phase distribution over one period with a circular normal function (von Mises function) $f(\theta) = r_{min} + (r_{max} - r_{min})\left(e^{k^2cos(\theta-\phi)} - e^{-k^2}\right)/\left(e^{k^2} - e^{-k^2}\right)$, where $r_{max}$ ($r_{min}$) is the maximal (minimal) firing rate, $\phi$ is the preferred phase shift of the cell and $k$ is a parameter measuring the concentration of spikes around the preferred phase ($k = 0$ indicates no modulation by phase, while a large $k$ indicates highly concentrated spikes). The phase shift was relative to the maximum MF stimulation at each cycle, and it could be positive, consistent with a delay of UBC maximum firing rate, or negative, consistent with a strong delay or an anticipation of phase. $CV_2$ is the mean of $2|ISI_1 - ISI_2|/|ISI_1 + ISI_2|$, where $ISI_1$ and $ISI_2$ are successive inter-spikes intervals (**Ruigrok et al., 2011**). Average ratio of depolarization / hyperpolarization and the steady-state firing rate in current-clamp protocols were estimated in the last second of 26 Hz steady stimulation. The peak of fast AMPA EPSC was obtained by fitting their decay with an exponential function and is given as the amplitude of the exponential decay. This measure avoids contamination by slow/buildup components of the response. For the same reason the charge of fast EPSCs is given as the product of the amplitude and of the decay time constant of the exponential fit. The buildup component of EPSC developed in the majority of the case with a delay of a few stimulations (3rd or 5th stimulation). We therefore distinguished the slow/steady-state component which follows fast EPSCs from the buildup component. The amplitude of the steady-state components was estimated as the amplitude of the base current of the exponential fit of the first EPSC. For comparison with other

components its charge was obtained by multiplying this amplitude by the duration of the stimulation (500 ms) and subtracted from the charge of the current envelope to yield the buildup charge.

Finally the rebound EPSC at the end of the stimulations was estimated by subtraction of an extrapolated exponential decay on which it superposes. The timecourse of this exponential decay was obtained by fitting the last 20% of the decay (amplitude wise), after the rebound, with an exponential function forced in amplitude and origin to the steady amplitude at the time when stimulation was stopped. Peak and $\tau_{ON}$ (rise time) were obtained from the exponential fit of the onset of the subtracted slow rebound EPSC. Statistical analysis was performed with Rstudio using Mann-Whitney U test, Friedman's test and Shapiro-Wilk test for normality of data. Results are quoted as mean ± SD unless stated otherwise. p<0.05 indicates statistical significance.

## Model of AMPAR-mediated synaptic currents in UBCs

UBC AMPAR-mediated synaptic currents exhibit complex dynamics including fast transients, slow buildup and post-stimulation slow rebound. We could quantitatively describe this synaptic dynamics with a linear combination of three types of AMPARs: 'close' receptors close to a release site, with instantaneous rise and exponential decay of the glutamate for the fast transients;; 'intermediate' receptors with a non-zero rise time for the rising of the rebound; 'far' receptors to give rise to the buildup and associated rebound.

AMPAR were modeled by a 4-state Markov model

$$C \underset{\beta_2}{\overset{\alpha_2[x]}{\rightleftharpoons}} O_2 \underset{\beta_1}{\overset{\alpha_1[x]}{\rightleftharpoons}} O_1 \underset{\beta_D}{\overset{\alpha_D}{\rightleftharpoons}} D,$$

with one closed state $C$, two open states $O_1$ and $O_2$, and a desensitized state $D$. Transitions between states were described by rate constants $\alpha$ and $\beta$. The transition rates from the close to the open state $O_1$, and from $O_1$ to $O_2$, were proportional to the glutamate concentration $x$. The fractions of receptors in each state ($r_2$ in $O_2$, $r_1$ in $O_1$, and $d$ in $D$) obey the following dynamics:

$$\begin{aligned} r_2' &= \alpha_2 x(1 - r_1 - r_2 - d) - (\beta_2 + \alpha_1 x)r_2 + \beta_1 r_1 \\ r_1' &= \alpha_1 x r_2 - (\beta_1 \alpha_D)r_1 + \beta_D d \\ d' &= \alpha_D r_1 - \beta_D d \end{aligned}$$

Each type of receptor is subjected to its own glutamate concentration dynamics $x$. The 'close' glutamate has the following dynamics:

$$dx/dt = x/\left[\tau_{decay}(1 + x/u)\right] + s\sum_k \delta(t - t_k),$$

At the spike time $t_k$, the glutamate concentrations increase by a step $s$, then decay with a concentration-dependent time constant $\tau_{decay}$ ($u = 30\ \mu M$). The concentration-dependence of the glutamate decay accounts for the non-exponential concentration decay found at proximity of the glutamate release sites, as a consequence of 2D or 3D diffusion (*Barbour, 2001*). Both 'intermediate' and 'far' glutamates with additional rise time to buildup have the following dynamics:

$$dx/dt = (y - x)/\tau_{decay}(1 + x/u),$$

$$dy/dt = -y/\tau_{rise} + s\sum_k \delta(t - t_k),$$

where the rise variable $y$ increases by a step $s$ when a spike occurs, and decays with a time constant $\tau_{rise}$.

The total synaptic current is

$$I_{AMPA} = \left[g_{close}\left(x_{close,1} + rx_{close,2}\right) + g_{int}\left(x_{int,1} + x_{int,2}\right) + g_{far}\left(x_{far,1} + x_{far,2}\right)\right]V_{hold},$$

with $V_{hold} = -60$ mV, and $g$s are conductance strengths for each receptor.

We fitted the recorded UBC synaptic traces with the model above. To reduce the number of parameters, we used the following values for the rate constants: $\alpha_1 = 0.03\ \mu M^{-1}\ ms^{-1}$, $\alpha_2 = 0.15\ \mu M^{-1}\ ms^{-1}$, $\alpha_D = 2\ ms^{-1}$, $\beta_1 = \beta_2 = 10\ ms^{-1}$ and fit other 11 parameters: $g$, $\tau_{decay}$, $s$, for all three types of receptors and $\tau_{rise}$ for both 'intermediate' and 'far' receptors.

The model was used to fit UBC synaptic currents triggered by a train of stimulation of 0.5 s at 50 Hz, close to the 1 Hz modulation parameters. After fitting, we fixed the parameters of each UBC, and then use these parameters to produce a simulated response to the modulated stimulation at 1 Hz used in the experiments for each UBC, as in *Figure 5*. We first calculated the correlation coefficients of the histograms of simulated EPSCs and experimental spike rate responses, and verified that most cells have a positive correlation, which implies that we emulated synaptic currents well enough to explain the spiking dynamics of most UBCs. We used the p-values to define well correlated cells in *Figure 5I*. We then fitted simulated EPSCs with a circular normal function to obtain the predicted phase shift of UBC spiking.

## Granule Cell Model

### Granule cells excitability and integration

Granule cells (GCs) were modeled as integrate-and-fire neurons, in which the membrane potential $V$ obeys the equation (when $V < V_T$)

$$CdV/dt = -g_L(V - E_K)exp(-(V - E_L)/5) - \bar{g}_{AHP} z_{AHP}(V - E_K) - I_{noise} - I_{inh} - g_{control}I_{syn}(t)$$

where $C$ = 4.9 pF, $g_L$ = 1.5 nS is the conductance of an inward rectifier potassium ($K_{IR}$) current, which is prominent in granule cells (*Rossi et al., 2006*). The activation curve of the $K_{IR}$ current was approximated by an exponential function, since the membrane potential of granule cells never visited potentials lower than the half $E_L$ = −90 mV. When the membrane potential reaches the threshold $V_T$ = −50 mV at the spike time $t_{spk}$, $V$ is set to 40 mV for a duration of the spike as $\tau_{dur}$ = 0.6 ms. The firing threshold $V_T$ is drawn randomly for each cell using a Gaussian distribution (−50 ± 2.5 mV; mean ± SD). After the spike, at $t = t_{spk} + \tau_{dur}$, the repolarizing potential is set to $V_{reset}$ = −65 mV, and an afterhyperpolarization (AHP) conductance ($\bar{g}_{AHP}$ = 1 nS, $E_K$ = −90 mV) is activated. The gating variable $z_{AHP}$ follows the dynamics $dz_{AHP}/dt = x_{AHP}(1 - z_{AHP}) - z_{AHP}/\tau_{AHP}$ where $\tau_{AHP}$=3 ms. The source variable $x_{AHP}$ obeyed the dynamics $dx_{AHP}/dt = -x_{AHP}/\tau_{AHP,x} + \delta(t - t_{spk} - \tau_{dur})$ where $\tau_{AHP,x}$=1 ms. The refractory period is set as $\tau_{ref}$=2 ms. Finally, a tonic inhibition current is added as $I_{inh} = g_{inh}(V - E_{CI})$ where $g_{inh}$ = 0.9nS, and $E_{CI}$ = −75mV. (*Mitchell and Silver, 2003*). In order to reproduce the low-frequency fluctuations of the cell membrane potential in the experimental data, a noise current $I_{noise} = g_N(V - V_E)$ with $V_E$=0 mV and a slowly fluctuating conductance $g_N$ is also taken into account. The conductance $g_N$ is described by an Ornstein-Uhlenbeck process $\tau_N dg_N/dt = -g_N + \sigma_N\sqrt{\tau_N}b(t)$, where $\sigma_N$=0.12 nS and $\tau_N$=1 s. $b(t)$ is a white-noise with unit variance density.

For a fair comparison between networks with or without UBCs, we control each GC's firing rate to be 5 Hz by multiplying a scaling factor $g_{control}$ to all incoming synapses, where $g_{control}$ was continually adjusted through the whole stimulation such that the mean firing rate to the desired value.

### GC excitatory synaptic currents

It has been shown that synaptic currents triggered by MF inputs onto GCs have three components: fast AMPA, slow AMPA and NMDA. All these synaptic currents were modeled using standard two-state kinetics. In addition, MF-GC synaptic dynamics are frequency-dependent (*Saviane and Silver, 2006*) and can be modeled with short-term dynamics, showing depression for the fast component and facilitation for the slow component.

We modeled the synaptic current as $I_x = g_x r(t)Y(V - E_x)$, where the scaling factor $Y$ is a voltage-dependent function for NMDA: $Y = 1/(1 + \exp(-(V - 84)/38)/ (\exp((V + 119)/38) + \exp(-(V + 45)/28)))$, (*Schwartz et al., 2012*) and $Y$ =1 for other receptors. $x$ indicates the receptor type ($AMPA_{fast}$, $AMPA_{slow}$ and NMDA). The gating variable $r$ is described by $r' = -r/\tau_{decay} - \alpha \cdot s(1 - r)$, with the rising variable $s$ as $s' = -s/\tau_{rise} + Ru \sum_k \delta(t \cdot t_{spk})$, where $Ru$ is the synaptic efficacy following the short-term plasticity modeled (*Izhikevich et al., 2003*; *Markram et al., 1998*) with the short-term depression variable $R$ recovering with time constant $\tau_{rec}$. $R' = (1 - R)/\tau_{rec} - uR\delta(t - t_n)$, and the short-term facilitation variable $s$ recorvering with time constant $\tau_{fac}$ $u' = (U - u)/\tau_{fac} + U(1 - u)\delta(t - t_n)$.

All synaptic parameters are listed in *Table 1*, in which the values are chosen to match experimental recorded EPSCs for both single stimulations and train stimulations (*Saviane and Silver, 2006*;

**Table 1.** Parameters of synapses

| Synapse | | Strength | EPSC kinetics | | | Short-term plasticity | | |
|---|---|---|---|---|---|---|---|---|
| Pre-Post | type | $g_{peak}$(nS) | $a$(1/ms) | $\tau_{rise}$(ms) | $\tau_{decay}$(ms) | $U$ | $\tau_{rec}$(ms) | $\tau_{fac}$(ms) |
| eMF-GC | AMPA$_{fast}$ | 0.4 | 3 | 0.3 | 0.8 | 0.5 | 600 | 600 |
| | AMPA$_{slow}$ | 0.8 | 0.3 | 0.5 | 5 | 0.5 | 600 | 600 |
| | NMDA | 0.96 | 0.35 | 8 | 30 | 0.05 | n.a. | n.a. |
| UBC-GC | AMPA$_{fast}$ | 1.6 | 3 | 0.3 | 0.8 | 0.5 | 12 | 12 |
| | AMPA$_{slow}$ | 3.2 | 0.3 | 0.5 | 5 | 0.5 | 12 | 12 |
| | NMDA | 3.84 | 0.35 | 8 | 30 | 0.05 | n.a. | n.a. |

*Schwartz et al., 2012*). The synaptic conductance $g_{peak}$ of eMF-GC and iMF(UBC)-GC is drawn randomly using a Gaussian distribution of SD 30% of the mean in *Table 1*.

## Granular layer network

We set up a simple network model to study the effects of neuronal and network mechanisms on the information transfer from the input, represented by the eMF activity, to the output, represented by the GC activity. The network includes 4500 GCs with 500 eMFs and 500 UBCs. Each GC receives 4 inputs, which are all eMF in the network with no UBCs, and eMFs and UBC in the network with UBCs. In networks with UBCs included, each synaptic input to GC could be either eMF or UBC with equal probability of 50%, as observed experimentally in organotypic cultures (*Nunzi and Mugnaini, 2000*) and as expected from the number (*Dugue et al., 2005*) and divergence of UBCs (*Berthie and Axelrad, 1994*).

Previous experimental studies have shown that the angular velocity of the rat head rotation is encoded linearly by the mossy fiber's firing rate (*Arenz et al., 2008*). Here we consider two types of eMFs, one that is in phase, the other in anti-phase with the velocity (*Arenz et al., 2008*). Therefore, we modeled the input velocity information as a sinusoid function of the eMF firing rate $\nu_{eMF} = r_{eMF}[1 + Asin(2\pi ft)]_+$, where $[.]_+$ rectifies the firing rate to be non-negative, $r_{eMF}$=26 Hz and the modulation $A = \frac{5}{3}fk$ obtained from in vivo data with movements at f=0.3 Hz (*Arenz et al., 2008*), and k is drawn uniformly within (0,1).

Similarly, we modeled the dynamics of UBC with a circular normal function with parameters $r_{max}$, $r_{min}$, $\phi$ and $k$ were obtained from the fitting of all 47 recorded UBCs in *Figure 6*. Half of the UBCs were modeled by sampling randomly from 47 recorded UBCs, and the other half was also sampled from recorded UBCs, but phase reversed (i.e. adding a phase shift of 180degrees), since eMFs can display phase and anti-phase behavior in vivo (*Arenz et al., 2008*). We also performed simulations where the parameters of all modeled UBCs were randomly chosen from the range of values of fitting parameters, to get more heterogeneous phase shifts in the firing rate curves. Results were indistinguishable from the results presented here.

With the given firing rate $\nu_{eMF}$ and $\nu_{UBC}$, we generated the spike trains by the time-rescaling method (*Brown et al., 2002*), then fed these input spike trains to GCs.

## Purkinje cell model

We modeled the Purkinje cell (PC) as a firing rate unit, whose firing rate was an instantaneous linear function of the weighted inputs from all GCs, i.e. $r_{PC}(t) = \sum i\, w_i r_i^{GC}(t)$, where $r_i^{GC}(t)$, is the average GCs firing rate at a given phase within an oscillatory cycle. We defined a 'target' firing rate of the PC, $r_{PC,target}(t) = \bar{r}(1 + \cos(2\pi ft - \phi))$, where $\phi$ is the preferred target phase to learn, and $\bar{r}$ is the average rate of 32 Hz (typical rate of PCs in vivo). To achieve this target firing rate, the synaptic weights of the Purkinje cell evolved according to the following learning rule: $\Delta w_i = \eta\, r_i^{GC}(t)E$, where $E = (r_{PC,target}(t) - r_{PC}(t))$ is the 'error signal', which could be implemented by climbing fibers. Weights $w_i$ were initialized as 0.5, and updated by $\Delta w_i$, and are set to zero if they become negative. The learning rate was $\eta = 0.001$. The learning process was simulated until convergence of both the weights and the error signal after 10000 cycles.

Codes for all numerical simulations are available on https://sites.google.com/site/jiankliu.

## Acknowledgements

This work has received support from CNRS, INSERM, Ecole Normale Supérieure, an ANR-BBSRC grant VESTICODE to NB and SD, fellowship from Ecole des Neurosciences de Paris (ENP) to PPM, and ANR-10-LABX-54 MEMO LIFE; ANR-11-IDEX-0001-02 PSL* Research University grants.

## Additional information

### Funding

| Funder | Grant reference number | Author |
|---|---|---|
| Centre National de la Recherche Scientifique | | Marco A Diana<br>Nicolas Brunel<br>Stéphane Dieudonné |
| Institut National de la Santé et de la Recherche Médicale | | Stéphane Dieudonné |
| Agence Nationale de la Recherche | ANR-BBSRC grant VESTICODE | Valeria Zampini<br>Jian K Liu<br>Marco A. Diana<br>Paloma P Maldonado<br>Nicolas Brunel<br>Stéphane Dieudonné |
| Agence Nationale de la Recherche | ANR-10-LABX-54 MEMO LIFE | Stéphane Dieudonné |
| Agence Nationale de la Recherche | ANR-11- 4 IDEX-0001-02 PSL* | Stéphane Dieudonné |

The funders had no role in study design, data collection and interpretation, or the decision to submit the work for publication.

### Author contributions

VZ, JKL, Acquisition of data, Analysis and interpretation of data, Drafting or revising the article; MAD, PPM, Acquisition of data, Analysis and interpretation of data; NB, SD, Conception and design, Analysis and interpretation of data, Drafting or revising the article

### Author ORCIDs

Nicolas Brunel, http://orcid.org/0000-0002-2272-3248

### Ethics

Animal experimentation: All animal manipulations were made in accordance with guidelines of the Centre national de la recherche scientifique. Protocols were approved under number 02235.02 of the general agreement C750520

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
