## [Decision Letter]

Thank you for submitting your article "Mechanisms and functional roles of glutamatergic synapse diversity in a cerebellar circuit" for consideration by *eLife*. Your article has been reviewed by two peer reviewers, and the evaluation has been overseen by a Reviewing Editor and Gary Westbrook as the Senior Editor. The reviewers have opted to remain anonymous.

The reviewers have discussed the reviews with one another and the Reviewing Editor has drafted this decision to help you prepare a revised submission.

Summary:

Longstanding models of vestibulo-ocular reflex adaptation have relied on a diversity of phases in granule cell responses to head velocity inputs, but it is not known whether such diversity actually exists or how it could arise. The present study examines the cellular mechanisms and the functional significance of signal transformation by unipolar brush cells (UBCs) in the vestibulocerebellum. The work shows that UBCs transform sinusoidally modulated (mossy fiber) synaptic inputs into diverse patterns of output firing and that differences in glutamate receptor expression and responses across UBCs accounts for the variation in temporal dynamics. A network simulation demonstrates that UBCs enable granule cells to fire at diverse phases in response to phase-restricted vestibular inputs. The work provides evidence that unipolar brush cells are plausible candidates for generating a diversity of granule cell responses in the vestibulocerebellum and the results are relevant to studies of the cerebellum and motor learning and also more generally, regarding in biophysical mechanisms for transforming stereotyped inputs into diverse outputs.

Essential revisions:

The reviewers were enthusiastic about the work and characterized it as "a very nice study" and said that it "provides several valuable insights to both synaptic physiologists and systems neuroscientists." They also said that the points were demonstrated "convincingly, with high quality recordings and clear figures." The questions that they had were fairly limited, and had to do with:

1) The extent of granule cell innervation by UBCs and, by extension, the accuracy of the model, and 2) the necessity of UBCs for cerebellar cortical processing. These points can probably be addressed with some additional modeling and rewriting.

Specifically:

1) In the network model is the assumption that GCs get half their inputs from UBCs realistic? If so, please provide references. If not, perhaps this ratio should be varied in the model. This seems an important point for constraining and interpreting the model results.

2) The final section of the Results, entitled "Purkinje cell learning of arbitrary input-output relationships depends on the presence of UBCs" overstates the role of UBCs in cerebellar signal processing, especially given that cerebellar circuits devoid of UBCs can "learn" arbitrary input-output relationships, e.g. for eyelid conditioning). Given this difference, please edit as appropriate the descriptions of the possible roles of UBCs and speculate on (or otherwise help readers think about) why UBCs might be highly expressed in vestibulo-cerebellum but not present in most parts of the cerebellum.

3) Regarding accessibility of the title and Abstract, please note that the phrase "private line transcoders" in the Abstract is unlikely to be readily understood by most readers. Please consider editing to a clearer phrasing. Likewise, "variegation" is not quite colloquial and usually refers to coloring (e.g. of leaves of plants). Possibly "variety" or "variable properties" can be substituted effectively.

---

## [Author Response]

*Essential revisions:*

*The reviewers were enthusiastic about the work and characterized it as "a very nice study" and said that it "provides several valuable insights to both synaptic physiologists and systems neuroscientists." They also said that the points were demonstrated "convincingly, with high quality recordings and clear figures." The questions that they had were fairly limited, and had to do with:*

1) The extent of granule cell innervation by UBCs and, by extension, the accuracy of the model, and […]

See response to point #1 below.

*2) The necessity of UBCs for cerebellar cortical processing. These points can probably be addressed with some additional modeling and rewriting.*

*Specifically:*

*1) In the network model is the assumption that GCs get half their inputs from UBCs realistic? If so, please provide references. If not, perhaps this ratio should be varied in the model. This seems an important point for constraining and interpreting the model results.*

Nunzi and Mugnaini (2000) estimated that approximately 50% of the MF endings in the granular layer of the developing nodulus are provided by axons of the UBCs, while the remaining 50% is coming from extrinsic MFs. However, the estimation of extrinsic MF terminals is based on organotypic cultures where glomeruli degenerating in the first two days were considered to be contributed by extrinsic MFs. The percentage of glomeruli formed by extrinsic MF could have been overestimated because some of the glomeruli generated by UBCs axons could also undergo degeneration in organotypic cultures.

A more realistic estimate was done by Dugué and colleagues (2005). They estimated with immunohistochemistry in fixed slices that 33% of the glomeruli in the nodulus contain one UBC dendritic brush. These include in principle extrinsic mossy fibers terminals onto UBCs and UBCs to UBCs terminals. Berthié and Axelrad (1994) counted on average two rosettes for each UBC, so each UBC is on average projecting two axonal endings. This means that 66% of the glomeruli in the nodulus contain a UBC terminal. Hence the 50% of intrinsic MF used in our modeling is a rather conservative figure.

*2) The final section of the Results, entitled "Purkinje cell learning of arbitrary input-output relationships depends on the presence of UBCs" overstates the role of UBCs in cerebellar signal processing, especially given that cerebellar circuits devoid of UBCs can "learn" arbitrary input-output relationships, e.g. for eyelid conditioning).*

*Given this difference, please edit as appropriate the descriptions of the possible roles of UBCs and speculate on (or otherwise help readers think about) why UBCs might be highly expressed in vestibulo-cerebellum but not present in most parts of the cerebellum.*

This issue is now addressed in a new paragraph in the Discussion. Delays seen in other systems or generated through other mechanisms are typically much shorter than the ones we show in this paper. Moreover, UBCs are also present in the hemispheres of the cerebellum and with increasing prevalence in higher mammals. It may therefore be argued that UBC function is rather general and that these cells are also involved in sensory-motor computations performed by the cerebellum other than vestibular.

*3) Regarding accessibility of the title and Abstract, please note that the phrase "private line transcoders" in the Abstract is unlikely to be readily understood by most readers. Please consider editing to a clearer phrasing. Likewise, "variegation" is not quite colloquial and usually refers to coloring (e.g. of leaves of plants). Possibly "variety" or "variable properties" can be substituted effectively.*

We have corrected these sentences accordingly.